# Invasomes and Nanostructured Lipid Carriers for Targeted Delivery of Ceftazidime Combined with N-Acetylcysteine: A Novel Approach to Treat *Pseudomonas aeruginosa*-Induced Keratitis

**DOI:** 10.3390/pharmaceutics17091184

**Published:** 2025-09-11

**Authors:** Mina Josef, Menna M. Abdellatif, Rehab Abdelmonem, Mohamed A. El-Nabarawi, Mahmoud Teaima, Hadeer M. Bedair, Alshaimaa Attia

**Affiliations:** 1Department of Industrial Pharmacy, College of Pharmaceutical Sciences and Drug Manufacturing, Misr University for Science and Technology, P.O. Box 77, Giza 12566, Egypt; mina.azmy@must.edu.eg (M.J.); elshimaa.mohamed@must.edu.eg (A.A.); 2Department of Pharmaceutics and Industrial Pharmacy, Faculty of Pharmacy, Cairo University, El-Kasr El-Aini Street, Cairo 11562, Egypt; mohamed.elnabarawi@pharma.cu.edu.eg (M.A.E.-N.); mahmoud.teaima@pharma.cu.edu.eg (M.T.); 3Department of Microbiology and Immunology, College of Pharmaceutical Sciences and Drug Manufacturing, Misr University for Science and Technology, P.O. Box 77, Giza 12566, Egypt; hadeer.bedair@must.edu.eg

**Keywords:** ceftazidime, *Pseudomonas aeruginosa*, antimicrobial resistance, bacterial keratitis, invasomes, nanostructured lipid carriers

## Abstract

**Objectives**: This study was designed to optimize a ceftazidime (CTZ)-loaded nanocarrier that could efficiently permeate across corneal tissues. Moreover, N-acetylcysteine (NAC) was combined with an optimized CTZ-loaded formula to augment the antimicrobial activity and facilitate the efficient healing of *Pseudomonas aeruginosa*-induced keratitis. **Methods**: Different CTZ-loaded invasomes (INVs) and CTZ-loaded nanostructured lipid carriers (NLC) were fabricated and fully characterized via the determination of the entrapment efficiency (EE%), particle size (PS), surface charge, and percentage of CTZ release. Next, NAC was added to the optimized formulae from each nanocarrier, which were further assessed through ex vivo corneal permeation and in vitro antimicrobial activity studies. Finally, an in vivo evaluation of the optimal nanocarrier in the presence of NAC was performed. **Results**: Both nanocarriers showed nanoscale PS with sufficient surface charges. CTZ-loaded NLC formulae showed a higher EE% range with a sustained drug release profile. Both optimized formulae showed a spherical shape and excellent stability. Moreover, the antibacterial activity and biofilm inhibition assessments confirmed the synergistic effects of NAC when combined with different CTZ-loaded nanocarriers. However, the optimized CTZ-loaded INV formula achieved higher corneal permeation and deposition compared to the optimized CTZ-loaded NLC formula. Finally, the in vivo assessment confirmed the dominance of the optimized CTZ-loaded INV formula combined with NAC, where the microbiological, histopathological, and immunohistopathological examinations showed the rapid eradication of keratitis. **Conclusions**: Recent strategies for the incorporation of antibiotics into nanocarriers, combined with mucolytic agents, can offer a promising platform to boost the therapeutic efficiency of antibiotics and prevent antimicrobial resistance.

## 1. Introduction

The advancement of antimicrobial resistance (AMR) poses a substantial risk to worldwide public health [1]. The extensive misuse and overuse of antibiotics have augmented the rise of resistant bacterial strains, causing once-effective treatments to become increasingly ineffective [2]. Globally, this concerning trend has resulted in prolonged illnesses, increased mortality rates, and greater healthcare burdens [3]. In 2013, the U.S. Centers for Disease Control and Prevention announced the beginning of the “post-antibiotic era”. In 2014, the World Health Organization (WHO) identified AMR as a serious worldwide health problem [4]. Although antimicrobial stewardship programs, regulatory policies, and healthcare provider education have demonstrated effectiveness [5,6], the overall impact on global antibiotic consumption is expected to increase by 200% by 2030 [7]. This highlights the crucial need for new strategies, including the advancement of novel antibiotics, alternative therapies, and enhanced diagnostic tools [2,8]. To guide global research and development efforts, the WHO recently revised its Bacterial Priority Pathogens List in 2024 [9]. This list classifies 24 antibiotic-resistant bacterial categories into three priority classes: critical, high, and medium [10].

One of the highest-priority pathogens is *Pseudomonas aeruginosa* (*P. aeruginosa*) [11], an opportunistic, Gram-negative bacterium associated with a broad spectrum of healthcare-associated infections, including pneumonia, bloodstream infections, and keratitis [11]. *P. aeruginosa* exhibits high intrinsic resistance and readily acquires further resistance mechanisms, making infections especially difficult to treat [12].

Because of the rapid progression and potential for immediate and irreversible vision loss, *P. aeruginosa*-caused eye infections, such as bacterial keratitis, are particularly concerning [13]. Bacterial keratitis is a severe infection of the corneal stroma that, if left untreated or improperly managed, can lead to corneal ulceration, scarring, and blindness [14]. Worldwide, microbial keratitis impacts more than two million individuals each year, with *P. aeruginosa* identified as one of the most commonly involved infections. The use of contact lenses is a prominent risk factor, as it encourages biofilm formation and corneal colonization by *P. aeruginosa*, therefore increasing the likelihood of infection [15]. The prevalence of *P. aeruginosa* as a causative organism in keratitis ranges widely, with reports of between 10% and 39% of all infectious keratitis cases and approximately 70% of contact lens-associated keratitis cases [16].

Treatment strategies typically include topical antibiotics; however, the growing incidence of multidrug-resistant *P. aeruginosa* strains significantly complicates clinical management [13]. Among the antibiotics utilized to treat *P. aeruginosa* keratitis, ceftazidime (CTZ), a third-generation cephalosporin, remains a cornerstone therapy [17,18]. Nevertheless, its effectiveness is increasingly compromised by resistance, highlighting the urgent need for adjunctive approaches that can restore or enhance antimicrobial efficacy [19,20,21,22,23]. This method enhances therapeutic outcomes by integrating non-antibiotic adjuvants with conventional antibiotics [24]. These adjuvants can disrupt bacterial biofilms, inhibit resistance processes, and enhance the efficacy of antibiotics [25,26]. N-acetylcysteine (NAC), an antioxidant and mucolytic agent, has recently garnered attention for its antibacterial and antibiofilm properties [27]. NAC may enhance overall antimicrobial activity by augmenting drug penetration, reducing bacterial adhesion, and destabilizing biofilm formation when used in conjunction with antibiotics like CTZ [28,29].

Ocular drug delivery faces substantial obstacles because of the complicated anatomy and various protective ocular barriers [30,31]. Traditional topical eye drops often result in minimal drug bioavailability, with less than 5% of the instilled dose being delivered to the target tissues [32]. This is primarily due to aspects such as tear drainage, blinking, and the occurrence of static barriers, including the cornea and conjunctiva, in addition to dynamic barriers, including tear flow and metabolic enzymes [33]. To address these drawbacks, innovative drug delivery systems are being fabricated, with nanocarriers showing great promise in enhancing drug solubility, improving bioavailability, and providing sustained release [34].

Nanocarriers have markedly enhanced ocular drug delivery by presenting advanced therapeutic strategies that exceed conventional drug delivery systems. These developments allow the penetration of corneal barriers, boost transcorneal drug permeability, prolong drug retention times, decrease the frequency of dosing, enhance patient adherence, offer controlled and prolonged release, and facilitate accurate drug targeting [35].

Currently, there are no widely commercially approved ocular products of CTZ due to its poor aqueous stability and rapid degradation in solution [36]. Only compounded formulations prepared by pharmacists on demand are used. Therefore, various studies have highlighted the possibility of using nanocarriers to enhance the delivery of CTZ to infected corneal tissue, prolong its corneal residence time, and overcome the stability limitations. Mohammadi et al. (2024) formulated mesoporous silica nanoparticles loaded with vancomycin and CTZ embedded in an in situ silk fibroin/sodium alginate composite hydrogel to treat bacterial keratitis. This complex delivery system showed prolonged release over 24 h for both drugs [37]. Moreover, Taghe and Mirzaeei (2024) fabricated a CTZ-loaded nanofiber as an ocular insert for the treatment of bacterial keratitis. These ocular inserts exhibited initial burst drug release for 12 h, after which they showed controlled release for 120 h [36].

The current study focused on utilizing simple and scalable nanocarriers that exhibit low ocular toxicity, enhanced corneal penetration capabilities, and controlled release properties. Therefore, two nanocarriers were selected: one consisted of invasomes (INVs), a deformable vesicular nanocarrier that may possess antimicrobial activity due to its terpenoid components, and the other was nanostructured lipid carriers (NLCs), which have low toxicity and improved stability and provide sustained release [38]. Hence, these systems were evaluated for their influences on the treatment of *P. aeruginosa* keratitis by boosting the ocular bioavailability of CTZ.

INVs are deformable vesicular systems that have garnered attention for their enhanced capabilities to penetrate biological barriers. While previous studies on INVs have primarily focused on transdermal drug delivery, their distinct compositions and characteristics suggest a potential application in ocular drug delivery, particularly for antimicrobial agents. INVs are typically composed of phospholipids, surfactants, and edge activators (often terpenes or terpene mixtures) [39]. The inclusion of terpenes is a key feature that distinguishes INVs from conventional liposomes and niosomes, as terpenes are known to enhance membrane fluidity and drug permeation across biological membranes, including the intricate corneal barriers [40]. This increased permeation enables higher drug concentrations to reach the target ocular tissues, thereby enhancing therapeutic outcomes. By facilitating the deeper and more efficient delivery of antimicrobial drugs, INVs can improve the efficiency of these drugs, potentially leading to the rapid eradication of infections and a reduction in treatment duration.

Additionally, the enhanced drug bioavailability accompanying INVs’ delivery can reduce the necessary dose frequency, which is especially beneficial in chronic or extended ocular treatments, thereby enhancing patient adherence. INVs exhibit advantageous stability characteristics, with some formulations remaining stable for extended periods even at room temperature, thereby facilitating their practical application in ophthalmic solutions [41]. Furthermore, they are suitable for sensitive eye tissues due to their low toxicity and minimal ocular discomfort, which is attributed to their composition, often based on biocompatible lipids [42]. Although the use of INVs in ocular antimicrobial therapy is still an emerging area, the principles underlying their enhanced permeation and stability, as demonstrated in transdermal applications, are directly applicable and highly relevant to ophthalmic drug delivery. Continued studies and optimization are crucial in fully harnessing the capabilities of INVs for targeted and effective ocular antimicrobial therapies.

NLCs are second-generation lipid nanoparticles that offer several merits over conventional drug delivery systems for ocular applications [43]. NLCs are a combination of solid lipids and oils, and this permits an increase in the drug-loading capacity and higher stability compared to solid lipid nanoparticles [44,45]. NLCs have emerged as potential solutions to enhance ocular drug delivery because of their capacity to enhance the solubilization of poorly soluble drugs, improve bioavailability, and enable controlled release, thereby decreasing the need for frequent administration, increasing patient compliance, and maintaining therapeutic levels over extended periods [33]. The lipidic nature and nanometric sizes of NLCs also promote enhanced residence on the ocular surface and increased permeation across barriers such as the cornea, facilitating drug delivery to the anterior and posterior segments of the eye [32]. NLCs consist of biocompatible and biodegradable lipids, which reduce eye discomfort and systemic toxicity, making them suitable for prolonged use [41]. This structural advantage, along with their versatility in being formulated as eye drops, gels, or ocular inserts, positions NLCs as a highly adaptable and efficient platform for ocular drug delivery [46]. Moreover, NLCs are readily scalable for industrial production, which facilitates their translation into clinical and commercial applications [47].

Given these considerations, this study investigates the development of an innovative ophthalmic formulation that combines CTZ-loaded INVs or CTZ-loaded NLCs with NAC for the healing of *P. aeruginosa*-induced keratitis. By leveraging the synergistic effects of combination therapy and advanced nanocarriers, this approach aims to overcome the current therapeutic limitations and offer a more effective strategy against resistant ocular infections.

## 2. Materials and Methods

### 2.1. Materials

Ceftazidime was generously gifted by the Egyptian International Pharmaceutical Industries Co. (Cairo, Egypt); N-acetylcysteine was generously gifted by South Egypt Drug Industries Co. (Cairo, Egypt); phosphatidylcholine from soybean, as well as fluorescein diacetate, was acquired from the Sigma Aldrich Chemical Co. (St. Louis, MO, USA); limonene (purity: 96%) was attained from Alfa Aesar (GmbH, Karlsruhe, Germany); chloroform and Tween 80 were obtained from El-Nasr Pharmaceutical Co. (Cairo, Egypt); glyceryl behenate (Compritol 888 ATO^®^), caprylocaproyl macrogol-8 glycerides (Labrasol^®^), and Lipophile WL 1349 (Labrafac™) were obtained as gifts from Gattefosse (St-Priest, France); ethanol and methanol were sourced from Merck (Darmstadt, Germany); caspase antibody (catalog no. A11953) was obtained from ABclonal (Woburn, MA, USA); and IL1β antibody (catalog no. GB12113) was obtained from Servicebio (Wuhan, China).

### 2.2. Fourier Transform Infrared Spectroscopy (FTIR)

FTIR spectroscopy was utilized to explore the possible chemical interactions and compatibility of CTZ and NAC within the formulation. The infrared spectra of the individual drugs and their physical combination (in a ratio of 1:1) were acquired using an FTIR device (Bruker, Coventry, UK). One milligram of each sample was blended with 100 mg KBr, compressed into a tablet, and the FTIR spectrum was recorded within the range of 4000–400 cm^−1^ at ambient temperature, with a resolution of 4 cm^−1^ and averaging over 16 scans [48,49].

### 2.3. Design of Experiment

The principal goal of the present design was to optimize a CTZ-loaded INV formulation according to specific conditions, such as achieving the highest entrapment efficiency (EE%), zeta potential (ZP), and percentage drug release after 8 h (Q8), while minimizing the particle size (PS) and polydispersity index (PDI). During primary screening, independent factors that could influence these parameters were identified, and a 2^3^-factor design employing Design Expert version 13 (Stat-Ease, Inc., Minneapolis, MN, USA) was employed to examine these independent variables and their interactions. The following factors were selected: limonene concentration (X_1_), lipid amount (X_2_), and terpene type (X_3_). The dependent variables EE% (Y_1_), PS (Y_2_), PDI (Y_3_), and ZP (Y_4_) were chosen, as illustrated in Table 1.

### 2.4. Preparation of CTZ-Loaded INVs

Initially, limonene and lipids were introduced into a round-bottom flask and solubilized in a mixture of organic solvents, i.e., 30 mL of 1:1 methanol to chloroform. A thin film was developed after evaporating the organic solvents at 40 °C under a vacuum for 30 min, employing a rotary evaporator (Heidolph VV 2000, Nuremberg, Germany) at 150 rpm [50]. After film formation, 10 mL of a 15% hydroalcoholic solution containing 10 mg of CTZ was added portion-wise to hydrate the film; then, the round flask was attached to the rotary evaporator for 15 min under the same conditions without a vacuum. The formed dispersion was then stored in a refrigerator overnight. Probe sonication was applied in 30 s on/30 s off cycles, not exceeding 25 °C [51].

### 2.5. Evaluation of CTZ-Loaded INVs

#### 2.5.1. Determination of Entrapment Efficiency

Briefly, two milliliters of each CTZ-loaded formula was introduced into semi-permeable membrane tubing (MW of 12,000–14,000 Da), subsequently submerged in 50 mL of distilled water, and stirred at 50 rpm for 10, 20, 30, 40, 50, and 60 min employing a magnetic stirrer. The free CTZ was diffused into the aqueous medium until a steady concentration was attained. The free CTZ was detected via a UV spectrophotometer (Shimadzu UV-1601 PC; Kyoto, Japan) at λ_max_ 253 nm. The EE% was estimated employing the following equation [52,53]:$$EE\left(\%\right)=\frac{Total\;CTZ\;amount-Diffused\;CTZ\;amount}{Total\;CTZ\;amount}\times 100$$

All evaluations were completed in triplicate ± SD.

#### 2.5.2. Determination of Particle Size, Polydispersity Index, and Zeta Potential

The PS, PDI, and ZP of the INV formulations were assessed by employing a Malvern Zetasizer (Malvern Instruments Ltd., Malvern, UK) through the dynamic light scattering method at ambient temperature, subsequent to appropriate dilution [53]. Each measurement was performed in triplicate.

#### 2.5.3. In Vitro Drug Release Study 

The in vitro drug release of CTZ from INV formulations was assessed utilizing a locally manufactured Franz diffusion cell with a functioning release area of 3.14 cm^2^ and a receiver cell volume of 50 mL (PBS, pH 7.4). The temperature of the receiving vehicle was set at 37 ± 1 °C and it was continuously agitated at 50 rpm by means of a magnetic stirrer [53]. A cellulose membrane (cutoff 12,000–14,000 Da) was positioned between the receiver and donor units. Two milliliters of the INV formula was placed in the donor compartment. One-milliliter samples were taken at hourly intervals and immediately analyzed via a UV spectrophotometer at 253 nm. In order to sustain sink conditions, an equivalent volume of fresh PBS was introduced into the receiver unit to substitute the withdrawn volume. The percentages of CTZ released were graphed against their corresponding time slots. The in vitro release parameters were computed; the steady-state flux (Jss) was determined as the slope of the linear segment, C_0_ represented the initial CTZ concentration, and the cumulative percentage of CTZ released after 8 h (Q8) was evaluated.

### 2.6. Formulation and Optimization of CTZ-Loaded NLCs

#### 2.6.1. Design of Experiment

The impacts of the total lipid concentration (X_1_), the percentage of solid lipids (X_2_), and the oil type (X_3_) on the characteristics of NLC formulae were assessed by employing a 2^3^ full factorial design. The examined responses were EE (Y_1_), PS (Y_2_), PDI (Y_3_), ZP (Y_4_), and Q8 (%), as illustrated in Table 2.

#### 2.6.2. Preparation of CTZ-Loaded NLCs

NLCs were formulated by employing a sequence of hot-melt emulsification, high-speed stirring, and ultrasonication. Briefly, Compritol and liquid oil were blended and warmed under adequate stirring at 85 °C to produce a homogenous, clear oil layer. The percentage of Compritol and the types of liquid oils are demonstrated in Table 2. For the aqueous phase, 10 mL of distilled water was prepared with a concentration of 2.5% *w*/*v* surfactant and included 10 mg of CTZ. Both phases were warmed individually for 10 min at 85 °C. Subsequently, the aqueous solution was dropped gradually into the oil phase and stirred for 15 min utilizing a magnetic stirrer at ambient temperature. The NLC dispersions were probe-sonicated for 10 min employing a VCX600 sonicator (Sonics and Materials, Newtown, CT, USA). The volume of each NLC formula was amended to 10 mL. The attained NLC formulae were maintained at 4  °C for additional characterization [54].

#### 2.6.3. Evaluation of CTZ-Loaded NLCs

As previously described in Section 2.5.

#### 2.6.4. Selecting the Optimized Formulae

The optimized INV formula and optimized NLC formula were selected utilizing the desirability function tool. The basis for choosing the formulation was to achieve the lowest PDI and PS, while maintaining the maximum Q8, EE%, and ZP values. To obtain a promising combined therapy for *P. aeruginosa*-induced keratitis, NAC was added to each optimized formula at a concentration of 75 mg/mL [29].

### 2.7. Characterization of Optimized Formulae

#### 2.7.1. Morphology

The morphologies of the chosen INV and NLC formulations were analyzed employing a transmission electron microscope (TEM) (JEOL, Tokyo, Japan). Each formulation was diluted tenfold and thereafter placed on a carbon-coated grid; for negative staining, one percent of phosphotungstic acid was utilized prior to drying for imaging [55].

#### 2.7.2. Differential Scanning Calorimetry (DSC)

To demonstrate the thermal characteristics of CTZ within INVs or NLCs, pure CTZ, lyophilized optimized CTZ-loaded INVs, lyophilized optimized CTZ-loaded NLCs, and individual components of each formula were analyzed using DSC. A thermal analyzer (TA-60, Shimadzu, Kyoto, Japan) was utilized to scan the DSC thermograms. The powders were tightly enclosed in aluminum pans, and their temperature was raised from room temperature to 400 °C at a constant rate of 10 °C/min.

#### 2.7.3. X-Ray Powder Diffraction (XRPD)

The optimized CTZ-loaded formulae were freeze-dried after the addition of 5% mannitol and frozen at −18 °C for 24 h. Next, the frozen solutions were kept in a freeze dryer (Labconco Corp., Kansas City, MO, USA) at −45 °C and a vacuum of 0.07 mbar for 48 h to obtain the lyophilized formulae [56,57]. The XRPD patterns of CTZ, NAC, and the lyophilized formulae were acquired using an XRD device (Rigaku, Tokyo, Japan) functioning at 35 kV and 15 mA, with a Cu Kα radiation source (λ = 1.25 Å). Wide-angle diffractions were scanned at a speed of 10°/min.

#### 2.7.4. Impact of Storage on Optimized Formulae

The ability of each nanocarrier to maintain its characteristics over 90 days was assessed. Each formulation was maintained at 4 °C for 90 days and then evaluated by matching its characteristics with those of the new formulation. Furthermore, each formulation was subjected to visual inspection for sedimentation.

### 2.8. Comparative Evaluation of Optimized CTZ-Loaded INV and Optimized CTZ-Loaded NLC Formulae

#### 2.8.1. Ex Vivo Corneal Permeation Studies

Ex vivo corneal permeation assessments were performed utilizing a Franz diffusion cell with an active diffusion area of 0.785 cm^2^. Freshly obtained cow corneal tissue was securely placed between the two units of the diffusion cell. Then, 2 mL of the optimized CTZ-loaded INV formulation or CTZ-loaded NLC formulation, corresponding to 2 mg of CTZ, was introduced into the donor unit. The receiver chamber was filled with 50 mL of PBS (pH 7.4) and heated at 37 ± 1 °C while being stirred magnetically at 50 rpm. At each hour, one milliliter of the permeation medium was collected, and an equivalent volume of fresh PBS was replenished in the receiver unit to ensure sink conditions. The samples were filtered and then assessed via the HPLC technique, where the mobile phase was composed of acetonitrile and buffer (pH 4) in a 40:60 (*v*/*v*) ratio. The Luna C18 column was utilized during the quantitative analysis of CTZ, where the flow rate was set to 1 mL/min and the retention time was 1.19 min [58]. The cumulative amount of CTZ permeating across the corneal tissues was graphed against time. The apparent corneal permeability coefficient (Kp) was calculated employing the following equation: Kp = Jss/C_0_. After finalizing the experiment, the corneal tissues were cleaned several times with distilled water to eliminate CTZ entirely. Then, they were soaked in 20 mL of distilled water and bath-sonicated for 15 min. Finally, samples were taken, filtered, and investigated using the previously validated HPLC technique to detect the amount of CTZ accumulated within the corneal tissue after 8 h.

#### 2.8.2. Confocal Laser Scanning Microscopy Study (CLSM)

Fluorescein diacetate (FDA) was utilized as a substitute for CTZ at a concentration of 1% *w*/*v* in the INV formula and NLC formula. A cow cornea was placed between the two units of the diffusion cell under identical settings as previously described in Section 2.8.1. The FDA-loaded INV and FDA-loaded NLC were applied to the cornea and left for 8 h. Then, the corneal tissues were cut into longitudinal sections and embedded in paraffin wax, sectioned, and inspected for fluorescence. The slides were captured by means of an LSM 710 inverted confocal microscope (Carl Zeiss, Oberkochen, Germany). The maximum wavelength of excitation of FDA was set at 497 nm, while emission was at 516 nm. The optical examination of the corneal tissues was performed using a 40× objective lens. CLSM images were attained using the LSM Image Browser software, version 4.2 (Carl Zeiss Microimaging GmbH, Jena, Germany). Moreover, the intensity of light was determined.

### 2.9. Microbiological Assessment of Optimized Formulae

#### 2.9.1. Screening of Antimicrobial Activity of Optimized Formulae

In the present research, the antimicrobial activity of the optimized IVS and NLC formulae was evaluated utilizing the well diffusion method, as depicted by Berghe and Vlietinck (1991) [59], against a clinical isolate of *P. aeruginosa* (MS99), which was sourced from the Misr University for Science and Technology (MUST) strain bank. Each formula was evaluated twice: the first was the CTZ-loaded formula, and the second was the CTZ-loaded formula combined with NAC. Furthermore, the inhibition zones of CTZ alone and in combination with NAC were also measured; both were dissolved in distilled water as solutions.

In this experiment, an adequate dilution (microbial suspension) was prepared, followed by the pouring of the medium into a Petri dish. Next, concentric 6 mm diameter wells were created, which were occupied with 20 µL of the formula or drug to be tested. After a 45 min pre-incubation period at 25 °C, the plates were incubated at 37 °C for 24 h to permit *P. aeruginosa* to grow. After the termination of the incubation period, the diameters of the zones of inhibition were determined. This technique guaranteed the radial diffusion of the antibacterial agent from the well, resulting in a clear and easily measurable zone of inhibition. The agent diffused radially and created a circular zone of inhibition on the previously inoculated agar surface with the *P. aeruginosa* suspension [60].

#### 2.9.2. In Vitro Antibacterial Activity of Optimized Formulae

##### Determination of MIC and MBC

The antimicrobial efficacy of the two formulations, the NLC and INV, against a clinical isolate of *P. aeruginosa* (MS99) was determined, conforming to the recommendations of the Clinical and Laboratory Standards Institute, utilizing the broth microdilution technique [61,62]. The minimum inhibitory concentration (MIC) and the minimum bactericidal concentration (MBC) were measured in two separate evaluations of each formula. The first evaluation involved testing the CTZ-loaded formula, while the second evaluation examined the CTZ-loaded formula combined with NAC. Additionally, the MIC and MBC of CTZ alone, as well as in combination with NAC, were also assessed. Both were dissolved in distilled water as solutions. Briefly, the MIC and MBC were assessed by the broth two-fold microdilution technique in Mueller-Hinton broth (MHB), as modified from the procedure described by Weinstein et al. (2020) [63].

In brief, 100 µL of sterile broth was instilled in all wells in the plate. Next, 100 µL of the antimicrobial agent was introduced to the first well of each row. Serial two-fold dilutions were created across the row (for example, from column 1 to column 10), ensuring that each transfer was mixed entirely to attain a range of concentrations. Then, 100 µL from the last well was removed to maintain equal volumes across all wells. A standardized inoculum of *P. aeruginosa* was prepared, typically at a concentration of 1 × 10^5^ CFU/mL. Then, 100 µL of the inoculum was introduced into each well, resulting in a final volume of 200 µL per well. For controls, the positive control consisted of wells containing *P. aeruginosa* but no antimicrobial agent (growth control). Negative control wells contained only broth (no inoculum or antimicrobial agent) [64]. To determine the MBC, an aliquot of 5 µL was withdrawn from the well that corresponded to the MIC, as well as from the wells with higher concentrations. The aliquots were subcultured onto Mueller-Hinton agar (MHA), and the plates were incubated at 37 °C for 24 h. The sample with the lowest concentration before the MIC, which showed no bacterial growth, was counted as the MBC. The results from three biological replicates, each with three technical wells, were averaged and are presented as the mean ± SD.

##### Biofilm Inhibition Assay

The inhibition of biofilm formation was evaluated utilizing flat-bottom 96-well plates, as stated previously by Haney et al. (2018, 2015) [65,66]. A clinical isolate of *P. aeruginosa* (MS99) was used as a model for a microorganism that forms biofilms. Briefly, 10 µL of *P. aeruginosa* suspension (final OD_600_ = 0.01) was introduced into the interior wells of a 96-well polystyrene microtiter plate containing 100 µL of the tested formulation. Control wells received the bacterial suspension without treatment. The plates were then incubated statically at 37 °C for the entire day to permit biofilm formation. After incubation, non-adherent cells were discarded by gently rinsing the wells three times with sterile PBS (pH 7.2). The remaining adherent biofilms were affixed utilizing 200 µL of methanol for 15 min, followed by air drying. Crystal violet solution at a concentration of 0.1% was used to stain the biofilms for 15–20 min at 25 °C. The excess stain was rinsed with distilled water, and the bound dye was dissolved utilizing either 95% ethanol or 30% acetic acid. The absorbance was then detected at 570 nm utilizing a microplate reader. The amount of biofilm inhibition was estimated relative to the amount of biofilm grown in the absence of the tested formulations (defined as 100% biofilm) and the media sterility control (defined as 0% biofilm). The biofilm inhibition % was estimated utilizing the following equation:Biofilm inhibition % = [OD Control − OD Test)/OD Control] × 100

The results from three biological replicates, each with three technical wells, were averaged and are presented as the mean ± SD.

##### Biofilm Detachment Assay

The ability of the tested formulations to eliminate already developed biofilms was assessed by the addition of 190 µL of MHB and 10 µL of inoculum to a polystyrene plate. After 24 h of *P. aeruginosa* biofilm formation at 37 °C in an aerobic atmosphere, the supernatant was discarded in a laminar flow hood to avoid disrupting the biofilm formed in the well. Subsequently, 100 µL of test material at varying concentrations was added to the wells containing mature biofilms, followed by incubation at 37 °C for an additional 24 h. After 24 h incubation, the eradication ability was determined using the crystal violet staining assay. All assays were conducted in triplicate [67]. The results from three biological replicates, each with three technical wells, were averaged and are presented as the mean ± SD.

### 2.10. In Vivo Assessment of Selected Optimized Formula

All animal care and studies conducted in this research were approved by the Research Ethics Committee of the Faculty of Pharmacy at Cairo University (Approval No. PI 3290), in compliance with the “Guide for the Care and Use of Laboratory Animals” set forth by the Institute of Laboratory Animal Research (Washington, DC, USA). Forty male Wistar albino rats without ocular damage or diseases, each weighing 150 g, were obtained from the MUST animal center to assess the therapeutic effects of the CTZ-loaded nanocarrier combined with NAC in a keratitis model induced by *P. aeruginosa*. The rats were maintained in a controlled environment with temperatures set between 24 °C and 26 °C and an established photoperiod of 12 h of light followed by 12 h of darkness for one week prior to the initiation of the study. They were individually housed in stainless-steel cages and provided with unlimited access to food and water.

The rats were randomly classified into five groups utilizing the random number table method, with eight rats in each group (*n* = 8). Group 1 (Gp1) was set as the negative control group and received neither corneal injury nor bacterial inoculation. All remaining rats were subjected to bacterial keratitis, where a corneal epithelial defect was created under anesthesia, and 0.05 mL of a suspension containing 1 × 10^8^ CFU/mL of the clinical isolate of *P. aeruginosa* (MS99) was instilled into the scarred right eye. Rats that developed keratitis by the third day post-inoculation were confirmed and included in the remaining groups. Only the right eyes of the animals were subjected to infection; the left eyes remained uninfected and served as internal controls. Group 2 (Gp2) was designated as the positive control group. No topical treatment was administered to Gp2 throughout the seven-day observation period. Group 3 (Gp3) received a combined topical treatment consisting of an aqueous solution of NAC (75 mg/mL) and CTZ (1 mg/mL). Group 4 (Gp4) was treated with an aqueous solution of CTZ (1 mg/mL). Group 5 (Gp5) received a CTZ-loaded nanocarrier (1 mg/mL) combined with NAC (75 mg/mL), while Group 6 (Gp6) received an aqueous solution of NAC (75 mg/mL). Ocular swabs were taken every 48 h from each rat and then suspended in 1 mL of sterile PBS and cultured to monitor the progression of the bacterial count throughout the treatment duration. At the end of the study (7 days after treatment initiation), the rats were sacrificed, and their globes were enucleated. These ocular tissues were subjected to microbiological, histopathological, and immunohistochemical investigation [68]. The investigators conducting group allocation and treatment administration were cognizant of the group assignments; however, those responsible for outcome evaluation and data analysis remained blinded to the group allocations during these stages of the experiment.

#### 2.10.1. Microbiological Assessment

To prepare for bacterial quantification, three rats from each group (*n* = 3) were selected, where each infected and treated eye was aseptically separated into anterior and posterior parts utilizing sterile scissors. The separated tissues were homogenized in 5 mL of 0.9% sterile saline. For CFU assessment, 10 µL of the homogenized corneal tissue was diluted 1:10,000 in sterile saline. Then, 100 µL of the diluted suspension was applied to nutrient agar plates and incubated at 37 °C for 24 h. Colonies ranging between 30 and 300 were counted, and the mean CFU/mL for each cornea was calculated [69]. The results were statistically assessed employing the Kruskal-Wallis test, followed by post hoc pairwise comparisons, using the Prism Instat software, version 8.

#### 2.10.2. Histopathology Study

After the termination of the in vivo experiment, three rats from each group (*n* = 3) were selected for histopathological analysis. Initially, the entire eye was preserved in a 10% *v*/*v* formalin saline solution. Following this, the corneal tissues were separated and then dehydrated with alcohol before being embedded in molten paraffin. Thin sections (2 μm) were subsequently prepared using a microtome, deparaffinized, and stained with hematoxylin and eosin. Ultimately, the stained sections were observed under a light microscope (DMS1000 B; Leica, Milton Keynes, UK) [70].

#### 2.10.3. Immunohistochemistry Assay

Paraffin sections from two rats from each group (*n* = 2) were affixed to positively charged slides utilizing the avidin–biotin–peroxidase complex (ABC) method. Sections were incubated with these antibodies. Then, the reagents needed for the ABC technique were incorporated (Vectastain ABC-HRP kit, Vector Laboratories, Burlingame, CA, USA). Marker expression was labeled with peroxidase and then colored with diaminobenzidine to differentiate the antigen–antibody complex. Negative controls were employed, employing non-immune serum instead of the primary or secondary antibodies. Immunohistochemistry-stained sections were observed utilizing an Olympus microscope (BX-63, Olympus, Tokyo, Japan). An analysis of the immunohistochemistry findings was performed by the estimation of the reaction area percentage in 7 microscopic fields, employing Image J 1.53t (Wayne Rasband and contributors, National Institutes of Health, Bethesda, MD, USA).

### 2.11. Statistical Analysis of Data

To evaluate the significant differences among the outcomes of the examined formulae, a one-way analysis of variance (ANOVA) test was utilized. The significance level was established at 0.05, and *p* < 0.05 was considered statistically significant.

## 3. Results and Discussion

### 3.1. Fourier Transform Infrared Spectroscopy

FTIR was employed to conduct compatibility studies, and the major peaks attributed to CTZ, corresponding to its functional groups, were observed at 3307–3404 cm^−1^ (N-H and N-H_2_ group axial deformations), 1750–1725 cm^−1^ (carboxylic acid C=O stretching), 1475–1600 cm^−1^ (aromatic ring C=C axial deformation), 1350–1300 cm^−1^ (C-N axial deformation), and 1680–1630 cm^−1^ (amide group C=O axial deformation). Similarly, for NAC, characteristic peaks were identified at 3200–3359 cm^−1^ (N-H stretching), 3200–3600 cm^−1^ (O-H stretching), 1571–1600 cm^−1^ (C=O stretching of CONH_2_), 1350–1310 cm^−1^ (C-N stretching), and 3000–2850 cm^−1^ (C-H stretching). These characteristic peaks were also detected in the physical mixture of CTZ and NAC, indicating that the individual functional groups were preserved and confirming that no chemical interaction had occurred between CTZ and NAC, as demonstrated in Figure 1 [71,72,73].

### 3.2. Factorial Design Optimization for INV Formulae

One of the objectives of this research was to improve the ocular bioavailability of CTZ. Therefore, the PS was considered the most crucial factor in the current study, as nanocarriers with a smaller PS can effectively penetrate corneal mucosal membranes and overcome ocular barriers, thereby facilitating deeper drug delivery into corneal tissues [74]. Therefore, a primary assessment was conducted to evaluate the impact of various variables on the PSs of the INVs. The independent variables that might impact the PS were selected, and their optimal ranges were determined. Moreover, limonene was selected as a terpenoid component in INVs due to its documented antibiofilm properties against *P. aeruginosa* [75,76].

The factors investigated for INVs were (X_1_) the limonene concentration, (X_2_) the lipid amount, and (X_3_) the sonication time. A total of 12 INV formulations were developed and evaluated to assess the effects of these factors on the EE% (Y_1_), PS (Y_2_), PDI (Y_3_), ZP (Y_4_), and Q8 (Y_5_), as presented in the Appendix A. The output of the design is demonstrated in Table 3.

### 3.3. Evaluation of CTZ-Loaded INVs

#### 3.3.1. Impacts of Formulation Variables on Entrapment Efficiency

The impacts of the studied factors, X_1_, X_2_, and X_3_, on the EE% of CTZ in the INV formulae are demonstrated in Table 3 and Figure 2. The EE% of CTZ in the INV formulae ranged from 64.31 ± 0.91% to 87.34 ± 0.52%. The results showed that increasing the limonene concentration (X_1_) substantially diminished the EE% (*p* = 0.0005). This finding may be ascribed to the fact that limonene, a highly lipophilic terpene (log P = 4.83), potentially caused the hydrophilic drug CTZ (log P = −0.78) to be expelled from the system. This behavior aligns with observations reported in the literature for hydrophilic drugs, where increasing the concentrations of lipophilic terpenes typically reduces the extent of the EE% [77].

Regarding the lipid amount (X_2_), it was observed that increasing the lipid amount resulted in a substantial improvement in the EE% (*p* < 0.0001). These outcomes may be associated with the fact that increased lipid amounts could endorse the formation of tight layers around INVs, thereby decreasing the leakage of CTZ [78]. Concerning the impact of the sonication time (X_3_) (*p* = 0.0047), as the sonication time was extended, the drug EE% declined due to CTZ leaking from the INVs [78]. The regression equation of the model was EE = +76.09 − 4.30 × A + 5.84 × B − 3.00 × C − 0.5687 × AB + 0.7313 × BC.

#### 3.3.2. Impacts of Formulation Variables on Particle Size

The optimum PS to prevent ocular irritation and maintain increased permeation through the corneal tissues is less than 200 nm [79,80,81]. The PSs of the INV formulations ranged from 43.50 ± 0.53 to 338.40 ± 3.5 nm. The ANOVA outcomes indicated that X_1_, X_2_, and X_3_ had a substantial impact on the PSs of the fabricated INVs. The limonene concentration (X_1_) had a direct impact on the PS (*p*  <  0.0001), where boosting the limonene concentration resulted in a larger PS. This effect is attributed to higher terpene concentrations inducing excessive membrane fluidity, which favors vesicle fusion and consequently increases the PS [41]. Regarding the lipid amount (X_2_), boosting the amount of lipids led to a significant increase in PS (*p* < 0.0001). These findings correspond with the EE% outcomes, where increased lipid amounts resulted in higher EE% values for CTZ in INVs. Concerning the consequences of prolonging the sonication time (X_3_) (*p* < 0.0001), the PS increased progressively as the sonication time increased. A longer sonication time was associated with an increase in PS, which is coherent with results reported in former studies, potentially due to particle re-agglomeration after initial de-agglomeration [82]. Additionally, the growth in PS may also be attributed to the expansion of the medium caused by elevated temperatures, which reduces shear energy and turbulence, thereby promoting the fusion of droplets [83]. The regression equation of the model was PS = +153.02 + 53.89 ×A + 56.49 × B + 33.55 × C + 25.48 × AB + 11.62 × AC + 1.27 × BC.

#### 3.3.3. Impacts of Formulation Variables on Polydispersity Index

The PDI values for all INV formulations ranged from 0.420 ± 0.010 to 0.598 ± 0.018, demonstrating acceptable size homogeneity. Upon the examination of the impact of the studied factors on the PDI (Y_3_), no statistically substantial model was found to fit the data.

#### 3.3.4. Impacts of Formulation Variables on Zeta Potential

The ZP values of all fabricated INV formulations revealed a negative charge and extended from 23.1 ± 0.77 to 40.1 ± 0.56 mV, as displayed in Figure 3. The ANOVA outcomes illustrated that only X_2_ (lipid amount) had a substantial effect on the ZP of the formulated INVs, with a *p*-value of <0.0001, where a higher lipid amount led to an increase in the ZP values. It was assumed that, in a medium of low ionic strength, the polar head group is oriented in such a way that the negatively charged phosphatidyl group is exposed outward. In contrast, the positively charged choline group is directed inward, leading to a net negative surface charge [77]. In contrast, X_1_ (limonene concentration) and X_3_ (sonication time) had *p*-values of 0.6572 and 0.2342, respectively. The regression equation of the model was ZP = +31.39 − 0.1500 × A. + 4.07 × B − 0.4250 × C + 2.53 × AB + 1.25 × BC.

#### 3.3.5. Impacts of Formulation Variables on In Vitro Drug Release

The in vitro CTZ release from various INV formulations was demonstrated in terms of Q8. The Q8 values of CTZ from the INV formulations ranged from 63.95 ± 1.1 to 92.21 ± 2.7%. The ANOVA outcomes indicated that X_2_ and X_3_ substantially affected the release of CTZ from INVs, with *p*-values of 0.0009 and <0.0001, respectively, while X_1_ had a non-significant impact, with a *p*-value of 0.1705. Regarding the lipid amount (X_2_), an inverse relationship was observed in Q8. This outcome could be related to the previously stated explanation, as an increase in lipid amount results in the formation of tighter layers around INVs, which may reduce CTZ leakage. For the sonication time (X_3_), an increase in Q8 was observed with increasing sonication times. This effect might be attributed to its negative impact on the drug EE%, which progressively declines due to heat generation and CTZ leaking from INVs, thereby facilitating faster drug diffusion and release. The regression equation of the model was Q8 = + 75.77 + 1.34× A − 4.60 × B + 6.51 × C − 1.11 × AB + 0.03 × BC.

### 3.4. Factorial Design Optimization for NLC Formulae

As previously mentioned, the PS may be considered the predominant factor that affects the amount of CTZ permeated across the corneal tissues. Therefore, a primary screening was conducted, and the factors that might affect the PSs of NLCs were included in the design. Furthermore, liquid oils with diverse HLB values were selected to assess their impacts on the characteristics of the fabricated NLC formulae. The variables investigated for NLCs were (X_1_) the lipid percentage, (X_2_) the solid lipid percentage, and (X_3_) the oil type. A total of twelve NLC formulations were prepared and evaluated to assess the consequences of these factors for the EE% (Y_1_), PS (Y_2_), PDI (Y_3_), ZP (Y_4_), and Q8% (Y_5_), as presented in the Appendix A. The output analysis of the design is illustrated in Table 4.

### 3.5. Evaluation of CTZ-Loaded NLCs

#### 3.5.1. Impacts of Formulation Variables on Entrapment Efficiency

The significance of the studied factors, X_1_, X_2_, and X_3_, on the EE% of CTZ in the NLC formulae is illustrated in Figure 4 and Figure 5. The EE% of CTZ-loaded NLC formulae extended from 82.17 ± 0.91 to 86.39 ± 0.8%, as displayed in Appendix A. The outcomes indicated that the total lipid concentration (X_1_) significantly decreased the EE% (*p* = 0.0056). This result is consistent with preceding studies, which have shown that increasing the total lipid percentage, comprising both solid and liquid lipids, in NLCs loaded with hydrophilic drugs significantly reduces the EE% while concurrently increasing the PS [84]. This behavior can be primarily attributed to the limited solubility of hydrophilic drugs in the lipid matrix, which favors the partitioning of drugs into the aqueous phase rather than their incorporation into the lipid core [85].

For increasing solid lipid percentages (X_2_), a substantial reduction in the EE% (*p* < 0.0006) was noticed. The EE% is significantly influenced by the drug’s solubility in the lipid matrix [86]. Since the drug used is hydrophilic, its compatibility with the lipophilic matrix is limited. The rise in the solid lipid ratio relative to liquid lipids can be attributed to the more ordered internal structure formed at higher lipid concentrations, which results in reduced aqueous compartments within NLCs [87]. Consequently, the system becomes less capable of encapsulating hydrophilic molecules [88].

Regarding the oil type, Labrafac, utilized as the liquid lipid component in NLC formulations, exhibits a comparatively low affinity for hydrophilic drugs owing to its non-polar triglyceride composition [89]. This restricts its capacity to retain hydrophilic molecules within the lipid matrix, resulting in drug diffusion into the adjacent aqueous medium during emulsification and subsequent particle solidification. Conversely, Labrasol is a non-ionic surfactant characterized by its amphiphilic properties [90] that has been shown to enhance the solubilization of hydrophilic drugs through hydrogen bonding and polar interactions. According to research, the polarity of Labrasol and its PEG chains is crucial to this mechanism [91]. Consequently, formulations using Labrasol exhibit markedly superior EE% values in comparison to those containing Labrafac. Moreover, Labrasol is significantly more hydrophilic (HLB 12) than the highly lipophilic Labrafac (HLB 5) [92,93]. This difference will affect their interactions with the hydrophilic drug CTZ. Statistical analysis supports this observation, with a reported *p*-value of <0.0001, confirming the significance of the variance. The regression equation of the model was EE% = +84.48 − 0.3925 × A − 0.6500 × B − 1.03 × C + 0.3000 ×AB + 0.1650 × AC − 0.6625 × BC.

#### 3.5.2. Impacts of Formulation Variables on Particle Size

The PS of an NLC is a critical feature that may influence its stability, bioavailability, and release profile and prevent local ocular irritation, while ensuring high absorption by the cornea [53,94,95]. The PSs of the NLC formulae extended from 138.7 ± 4.45 to 262.4 ± 7.35 nm. The ANOVA outcomes indicated that X_1_ and X_2_ substantially impacted the PSs of the fabricated NLCs.

The total lipid concentration (X_1_) significantly affected the PS (*p* = 0.0015), with an increase in the total lipid percentage resulting in a larger PS. This finding aligns with the prior literature, indicating that elevated lipid concentrations may diminish the available area for lipid dispersion in the medium, thereby leading to an increase in PS. Furthermore, as the lipid concentration increases, the aggregation of lipid molecules becomes more pronounced, aiding the formation of larger particles [55,96].

Increasing the solid lipid percentage (X_2_) led to the enlargement of the PS of the NLC (*p* = 0.0006). These observations align with findings from previous studies, as higher solid lipid concentrations increase the PS and lead to agglomeration during preparation, resulting in larger particles. Conversely, higher liquid lipid concentrations promote the formation of smaller, more uniform particles [86]. This was also observed in another study, where the percentage of solid lipids was adjusted to attain the target PS of an NLC for specific applications. In contrast, an increased percentage of liquid lipids decreased the viscosity and surface tension of NLCs [97]. Finally, the oil type (X_3_) demonstrated a non-significant difference, with a *p*-value of 0.1456. The regression equation of the model was PS = +199.44 + 14.39 × A + 17.49 × B + 3.23 × C + 3.49 × AB − 9.69 × AC − 20.29 × BC.

#### 3.5.3. Impacts of Formulation Variables on Polydispersity Index

The PDI values for all NLC formulations ranged from 0.287 ± 0.013 to 0.511 ± 0.022, demonstrating the acceptable homogeneity of the NLC formulations. Upon investigating the impacts of the studied factors on the PDI (Y_3_), no significant model fit was found for the data [98].

#### 3.5.4. Impacts of Formulation Variables on Zeta Potential

The ZP values of the NLC formulae extended from 17.22 ± 1.6 to 29.27 ± 0.98 mV. These values are acceptable, as ZP values around ±20 mV are considered acceptable [54,99]. All formulated NLCs had a negative surface charge, owing to the acidic groups in solid and liquid lipids [54].

The ANOVA results showed that only X_2_ notably affected the surface charges of the formulated NLCs, with *p*-values of 0.0001. The solid lipid percentage (X_2_) showed a substantial influence on the ZP values. As the solid lipid percentage increased, the ZP value also increased. It was expected that a higher solid lipid ratio would enhance the organization of the lipid matrix, potentially exposing more charged groups (e.g., fatty acid residues of Compritol) on the NLC surface. This increases the electrostatic repulsion between particles, reflected in a higher ZP value [44,99].

Finally, the total lipid concentration (X_1_) and oil type (X_3_) showed no significant differences, with *p*-values of 0.1054 and 0.3446, respectively [54]. The regression equation of the fitted model was ZP = +21.53 + 0.1950 × A + 1.07 × B + 0.0842 × C + 2.89 × AB − 1.65 × AC − 2.16 × BC.

#### 3.5.5. Impacts of Formulation Variables on In Vitro Drug Release

The in vitro drug release from different NLC formulae was demonstrated in terms of the cumulative percentage of drug released after 8 h (Q8%). The Q8 of CTZ from NLC formulae ranged from 61.68 ± 0.8 to 79.52 ± 1.9%, as demonstrated in Appendix A. The ANOVA outcomes showed that X_2_ and X_3_ significantly impacted the release of the formulated NLC, with *p*-values of 0.0001 and 0.0008, respectively. Regarding the lipid composition percentage (X_2_), a direct positive relationship was observed with in vitro drug release. This result could be related to the formerly stated explanation, as an increase in the solid lipid ratio relative to liquid lipids results in a more ordered and tightly packed lipid matrix [100], which might enhance the release of CTZ from the system.

For (X_3_) the oil type, Labrafac exhibited superior drug release compared to Labrasol, as Labrafac is a liquid lipid characterized by a lower melting point and higher fluidity relative to Labrasol [93]. This created a more disordered lipid matrix in the NLC, promoting the faster diffusion and release of CTZ. The imperfect crystalline structure formed by Labrafac permitted CTZ to partition more readily from the lipid matrix into the aqueous release medium, resulting in a higher release rate [101]. The regression equation of the fitted model was Q8 = +69.32 + 0.3550 × A + 4.90 × B + 2.76 × C − 0.5168 × AB + 1.36 × AC + 0.7432 × BC.

### 3.6. Selection of Optimized Formulae

To select the optimized INV formula and optimized NLC formula, definite constraints were adjusted in Design-Expert. These requirements were applied to identify the formulae with the highest EE%, ZP, and Q8, as well as the lowest PS and PDI. The optimized IVN formulation that fulfilled these conditions consisted of a 0.129% limonene concentration, 144.865 mg of lipids, and a sonication time of 10 min. The calculated desirability score of the optimized formulation was 0.730.

In comparison, the optimized NLC formulation that fulfilled these parameters was F6, composed of 2% total lipids, 90% solid lipids, and Labrasol as the oil phase. The calculated desirability score of the optimized formulation was 0.618. Consequently, these formulae were chosen for further evaluation.

### 3.7. Characterization of Optimized Formulae

#### 3.7.1. Morphology

The TEM images of the optimized formulae showed that both nanocarriers were spherical, as revealed in Figure 6. Additionally, the PS result obtained using Zetasizer aligned with the TEM examination.

#### 3.7.2. Differential Scanning Calorimetry

DSC was utilized to study the crystalline states and thermal behavior of the individual components and the optimized formulae. Regarding the optimized INV formula, DSC thermograms were recorded for CTZ, NAC, lipids, limonene, and the optimized INV formula, as demonstrated in Figure 7a. The thermogram of CTZ exhibited a sharp endothermic peak at 160.16 °C, confirming its crystalline state. NAC exhibited a characteristic peak at 118.36 °C. Lipids exhibited multiple transitions, typically at 72.82 °C (gel-to-liquid crystalline phase). Limonene, a volatile terpene, did not present a distinct melting peak in DSC due to its low melting point (−74 °C) and high volatility (boiling point ~178 °C, flash point 48 °C); only a small peak at 191.11 °C was observed. In contrast, the thermogram of the optimized INV formulation did not show any endothermic peak corresponding to CTZ, suggesting that it was successfully loaded within the phospholipid bilayer and potentially transformed into an amorphous or molecularly dispersed state [51]. This thermal behavior supports the efficient entrapment of CTZ within the INV vesicles, which may enhance its stability and bioavailability for ocular drug delivery applications.

Regarding the optimized NLC formula (Figure 7b), Compritol displayed a characteristic melting peak at 73.67 °C, while Tween 80 showed peaks at 108.2 °C and 132.32 °C. Labrasol showed a minor peak at 151 °C. In contrast, the thermogram of the optimized NLC formulation did not reveal any endothermic peak corresponding to CTZ, suggesting its successful encapsulation within the lipid matrix [54,55].

#### 3.7.3. X-Ray Powder Diffraction (XRPD)

Pure CTZ exhibited peaks at 10.21°, 14.95°, 16.42°, 18.88°, 24.04°, 26.87°, and 30.14°, while NAC exhibited peaks at 16.13°, 21.67 °, 23.35°, 30.55°, 32.05°, and 35.63°, which were indicative of their crystalline states. When evaluating the XPRD patterns of the lyophilized formulae, no peaks related to CTZ were detected, indicating that CTZ was loaded into the nanocarrier in the amorphous state. Moreover, no peaks related to NAC were observed, as illustrated in Figure 8.

#### 3.7.4. Impact of Storage on Optimized Formulae

The impact of storage on the selected formulae was examined. Visual examination revealed no sedimentation or vesicle aggregation throughout the storage time in both formulae. Additionally, the measurements of the EE%, PS, PDI, and ZP were 86.07 ± 2.28%, 112.9 ± 4.5 nm, 0.506 ± 0.06, and −33.7 ± 0.90 mV, respectively, for the optimized INV formula. In comparison, the optimized NLC formula yielded values of 82.09 ± 1.6%, 218 ± 5.6 nm, 0.280 ± 0.019, and 19.5 ± 2.4 mV, respectively. These outcomes demonstrated no statistically significant difference compared to the newly prepared formulae (paired *t*-test, *p* > 0.05). These results confirmed the ability of both nanocarriers to maintain their physicochemical features throughout the storage period.

### 3.8. Comparative Evaluation of Optimized CTZ-Loaded INV and Optimized CTZ-Loaded NLC Formulae

#### 3.8.1. Ex Vivo Corneal Permeation Studies

The permeation studies demonstrated enhancements in the cumulative permeated amount of CTZ from either INVs or NLCs compared to the drug solution, as demonstrated in Figure 9. The cumulative amount permeated across the corneal tissue was 970.85 ± 0.97, 786.78 ± 2.39, and 417.67 ± 0.88 µg/cm^2^ for the CTZ-loaded INVs, CTZ-loaded NLC, and CTZ solution, respectively. The Kp values of the CTZ-loaded INVs, CTZ-loaded NLC, and CTZ solution were 0.098, 0.080, and 0.037 cm/h, respectively. The improvement in CTZ corneal permeation from the nanoformulations might be attributed to the smaller PS, which led to a higher interfacial area accessible for drug exchange, consequently facilitating drug delivery through the hydrated network of the corneal stroma and accordingly enhancing drug efficacy.

Compared to the CTZ solution, CTZ-INVs demonstrated a nearly 2.6-fold increase and a 1.2-fold increase in Kp compared to the drug solution and NLC formula, respectively. Furthermore, the corneal deposition achieved by CTZ-INVs was 105 µg/cm^2^, while the CTZ-loaded NLC and CTZ solution attained values of 66.87 µg/cm^2^ and 30.39 µg/cm^2^, respectively. The significantly higher corneal permeation and deposition achieved by the optimized INV formulation could be due to the presence of terpenes, which may have led to disturbances in the tight junctions of the corneal epithelial cells. Additionally, the ultra-deformability of INVs enabled the vesicles to have greater membrane flexibility, allowing them to efficiently penetrate the cornea [102].

#### 3.8.2. Confocal Laser Scanning Microscopy Study

CLSM assessed the efficiency of permeation and the fluorescence light intensities of FDA-labeled INVs, FDA-labeled NLCs, and an FDA solution. Images obtained with CLSM showed that the FDA-labeled INVs and FDA-labeled NLCs exhibited higher fluorescence deposition in various corneal tissues compared to the FDA solution (Figure 10). Both nanocarriers displayed homogeneous diffusion; however, when computing the maximum intensity of fluorescent light, there was substantial (*p* ≤ 0.05) variance among the various formulations and the FDA solution, as the mean intensity was 97.31 ± 0.22, 57.97 ± 0.97, and 23.33 ± 0.18 for the FDA-labeled INVs, FDA-labeled NLCs, and FDA solution, respectively. The CLSM outcomes confirmed the ex vivo permeation outcomes, where the INV formula had higher diffusion capabilities than the CTZ solution or CTZ-loaded NLCs.

### 3.9. Microbiological Assessment of Optimized Formulae

#### 3.9.1. Determination of MIC and MBC

The results regarding the MIC and MBC are presented in Figure 11. The results showed a promising synergistic effect when NAC was added to CTZ. While the MIC of CTZ alone in distilled water was 8 μg/mL and the MBC was 16 μg/mL, the addition of NAC reduced both the MIC and MBC to 4 and 8 μg/mL, respectively. Furthermore, both tested nanocaarier formulations, i.e., the NLC and INVs, enhanced the antimicrobial activity of CTZ, as evidenced by the reductions in the MIC and MBC values of CTZ to 4 and 8 μg/mL, respectively, when CTZ was loaded into the tested formulations, compared to CTZ alone in distilled water. Interestingly, both the CTZ-loaded INVs and CTZ-loaded NLC in the presence of NAC reduced the MIC and MBC to 1 and 2 μg/mL, respectively. The MIC and MBC values of CTZ alone were notably reduced upon the addition of NAC, indicating a synergistic interaction. This is consistent with recent studies demonstrating that NAC can potentiate the activity of β-lactam antibiotics, including CTZ, against resistant bacterial strains by disrupting biofilms and enhancing antibiotic penetration [28]. Furthermore, the incorporation of CTZ into the NLC and INV formulations further augmented its antimicrobial activity, as evidenced by the reductions in the MIC and MBC values. Nanocarrier systems have been demonstrated to enhance drug delivery, stability, and sustained release, thereby increasing the local concentrations of antibiotics at the site of infection and improving their efficacy [103,104].

#### 3.9.2. Determination of Biofilm Inhibition Assay

The anti-biofilm activity of the tested formulations, loaded with CTZ and in the presence of NAC, was compared to that of the aqueous solution of CTZ alone and the aqueous solution of CTZ combined with NAC at concentrations below the determined MIC (0.5–0.25 MIC). At all investigated concentrations, the tested formulations (CTZ-loaded INVs, CTZ-loaded NLC, CTZ-loaded INVs combined with NAC, CTZ-loaded NLC combined with NAC, CTZ alone, and aqueous solution of CTZ and NAC) inhibited the formation of *P. aeruginosa* biofilms, with inhibition ranging from 33% to 40% at sub-MIC concentrations. Interestingly, the CTZ-loaded INVs combined with NAC and the CTZ-loaded NLC combined with NAC formulations exhibited the highest biofilm inhibition activity (40%) compared to other formulations, the drug alone, or the drug combined with NAC, as shown in Figure 12. These findings are consistent with recent studies that highlight the challenges posed by biofilm-associated infections and the need for innovative strategies to disrupt biofilm formation and enhance antibiotic efficacy [103,104]. NAC is well documented for its mucolytic and biofilm-disrupting properties, which facilitate antibiotic penetration and potentiate antimicrobial activity [28]. The observed synergy between NAC and CTZ, especially when delivered via INVs or NLCs, may be ascribed to the combined effects of enhanced drug delivery, increased bacterial membrane permeability, and disruption of the extracellular polymeric substance (EPS) matrices of biofilms. Nanocarrier systems, such as NLCs and INVs, have been demonstrated to enhance the delivery and sustained release of antibiotics, leading to higher local drug concentrations and prolonged interactions with biofilm-embedded bacteria. Recent progress in biomimetic nanocarriers further highlights their roles in targeting biofilm environments and overcoming the protective barriers that limit the effectiveness of traditional antibiotics.

#### 3.9.3. Determination of Biofilm Detachment Activity

The investigated formulations substantially detached the previously formed *P. aeruginosa* biofilms at sub-MIC levels when compared to the control, untreated wells (one-way ANOVA, Tukey’s post hoc test, *p* < 0.0001) (Figure 13). Remarkably, the tested formulations exhibited excellent biofilm detachment activity against *P. aeruginosa*, with a percentage of 97%. Recent studies have emphasized the difficulty in eradicating mature *P. aeruginosa* biofilms due to their dense EPS matrices, which impede antibiotic penetration and protect embedded bacteria. NAC has been shown to disrupt the biofilm structure by breaking disulfide bonds within the EPS, thereby enhancing the susceptibility of bacteria embedded in biofilms to antibiotics [28]. The synergistic effect detected in this study may be attributed to the combined action of NAC’s mucolytic properties and the improved delivery and sustained release of CTZ via nanocarriers.

Nanocarriers have gained consideration for their ability to enhance the delivery of antimicrobial agents to biofilm sites, prolong drug retention times, and facilitate deeper penetration into the biofilm matrix [103]. The high detachment percentage observed in this study aligns with recent reports demonstrating the efficacy of nanocarrier-based systems in disrupting established biofilms and reducing bacterial loads.

### 3.10. In Vivo Assessment of Selected Optimized INV Formula

Based on the ex vivo studies and microbiological assessments of the optimized formulae, one of the two noncarriers was selected for in vivo assessment. Although both formulae, in the presence of NAC, reduced the MIC and MBC to 1 and 2 μg/mL, respectively, as compared to 8 and 16 μg/mL exhibited by the drug solution, the optimized CTZ-loaded INVs exhibited a larger amount of drug permeated and deposited across the corneal tissue (970.85 ± 0.97 and 105 µg/cm^2^), compared to those exhibited by the CTZ-loaded NLC (786.78 ± 2.39 and 66.87 µg/cm^2^), respectively. Therefore, the optimized CTZ-loaded INVs combined with NAC were selected for in vivo assessment.

#### 3.10.1. Microbiological Assessment

The baseline microbial count in the infected groups was 3 × 10^5^ ± 0.2 CFU/mL before treatment. The count increased to 10 × 10^7^ ± 0.54 CFU/mL in the positive group, while, in the groups treated with the aqueous CTZ solution, the aqueous solution of NAC, and the aqueous solution of CTZ and NAC, the burden decreased to 4 × 10^3^ ± 0.9, 8 × 10^5^ ± 1.6, and 1.2 × 10^2^ ± 0.78 CFU/mL, respectively. A substantial decrease in the microbial count was observed in the group treated with CTZ-loaded INVs combined with NAC, reaching 7.0 × 10 ± 0.67 CFU/mL, which demonstrates a log reduction of nearly 7.15, representing the almost complete eradication of the bacterial count compared to the positive group (*p* = 0.0102). The enhancement in the eradication of *P. aeruginosa* corneal infection achieved by CTZ-loaded IVNs combined with NAC might be due to the anti-pseudomonal activity of limonene, together with the enhanced diffusion of INV vesicles across the corneal tissue and the synergistic antibacterial activity exhibited by combining NAC with CTZ.

#### 3.10.2. Histopathology Study

The histopathological examination of the corneal tissue showed that the cornea in Gp1 exhibited a normal histological structure, including the epithelium and stroma, while the positive group, GP2, demonstrated the complete sloughing of the corneal epithelium, with the infiltration of the stroma by a high number of inflammatory cells—mainly neutrophils and lymphocytes (Figure 14). Gp3, which was treated with an aqueous solution of NAC and CTZ, showed the congestion of newly formed blood vessels, with the infiltration of the stroma by a few mononuclear inflammatory cells. Gp4, treated with CTZ solution, showed mild stromal edema with infiltration by a moderate number of inflammatory cells—mainly lymphocytes and eosinophils—in addition to the congestion of newly formed blood vessels. Gp6, treated with CTZ-loaded INVs in the presence of NAC, demonstrated a cornea with a normal histological structure, including the epithelium and stroma. Finally, Gp6, treated with an aqueous solution of NAC, revealed a severe hemorrhage in the corneal epithelium and stroma, with the infiltration of the stroma by a moderate number of inflammatory cells—mainly lymphocytes and eosinophils. These results confirmed the microbiological assessment, where the group treated with the CTZ-loaded INV formula showed the almost complete eradication of the microbial infection. On the other hand, Gp6, treated with an aqueous solution of NAC, showed signs of existing keratitis, because NAC alone may have relatively weak direct antibacterial effects, but it gains significance through biofilm inhibition and antibiotic synergy. This suggests that NAC may serve as an adjuvant therapy to enhance antibiotic treatment and help to combat antibiotic resistance.

#### 3.10.3. Immunohistochemistry Assay

The immunohistochemistry assay demonstrated findings that confirmed those of the histopathological examination, where the negative control (Gp1) showed the negative expression of caspase 3 and IL1β in the corneal epithelium and stroma. In contrast, the positive group (Gp2) showed strong positive expression for caspase 3 and IL1β in the corneal epithelium and stroma. Gp3 and Gp4 showed mild positive expression for caspase 3 and IL1 β in the corneal epithelium and stroma. Gp5 demonstrated findings similar to Gp1, showing the negative expression of caspase 3 and IL1β in the corneal epithelium and stroma. Finally, Gp6 showed, to some extent, strong positive expression for caspase 3 and IL1β in the corneal epithelium and stroma, as illustrated in Figure 15 and Figure 16.

In the current study, two simple and scalable nanocarriers were investigated and optimized to select the optimal nanocarrier for the delivery of CTZ across corneal tissues to treat keratitis induced by *P. aeruginosa.* The optimized CTZ-loaded INVs exhibited a PS below 200 nm with an acceptable surface charge (above 30 mV). In comparison, the optimized CTZ-loaded NLCs exhibited a larger PS and lower ZP values. The NAC was combined with the optimized formula from each nanocarrier system. Both optimized formulae, in the presence of NAC, maintained their characteristics throughout the storage period and reduced the MIC and MBC compared to the pure CTZ solution. The optimized CTZ-loaded INVs exhibited a larger amount of drug permeated and deposited across the corneal tissue compared to the CTZ-loaded NLC. Therefore, the INV formula was selected as the optimal nanocarrier for the loading of CTZ in in vivo studies, which demonstrated the efficacy of combining this formula with NAC in eradicating bacterial infections and restoring the normal histopathological features of infected corneal tissue.

Potential limitations of the current research work include the inherent variability in the clinical isolates of *P. aeruginosa*, which may result in variations in antimicrobial susceptibility [105]. The observed synergistic antibacterial activity of CTZ-loaded nanocarriers in combination with NAC may not be generalized due to this variability, and, as a result, extensive microbiological screening is required. Although the optimized CTZ-loaded INVs, in combination with NAC, demonstrated efficacy in the in vivo model, their performance in human ocular tissues must be carefully assessed, because animal models may not entirely resemble the human ocular environment. The physiological and anatomical differences between animal models and human subjects, as well as patient variability in corneal permeability and the immune response, may also affect the outcomes. Although NAC shows low ocular toxicity, and a low concentration of limonene was utilized in the INV formulation (0.1−0.3%), long-term safety remains a critical consideration [106]. Additionally, translation challenges from laboratory formulations to clinical use include the scalability of nanocarrier production, the reproducibility of physicochemical properties, and regulatory difficulties associated with novel nano-based ocular therapies [107]. These factors underscore the importance of conducting thorough clinical trials to confirm the safety, effectiveness, and optimal dosage schedules in human participants, as well as the ideal dosing regimens.

## 4. Conclusions

In the current study, two different nanocarriers, namely invasomes and nanostructured lipid carriers, have been developed and optimized to achieve an optimal formula for the incorporation of ceftazidime. Both systems showed acceptable features in terms of entrapment efficiency, particle size, homogeneity, and stability. Furthermore, both formulae exhibited enhanced antibacterial activity, especially in the presence of N-acetylcysteine; however, the invasomal formula demonstrated higher permeation capabilities, as confirmed by confocal laser scanning microscopy. Therefore, the ceftazidime-loaded invasomal formula combined with N-acetylcysteine demonstrated superior antibacterial activity throughout the in vivo studies. These results highlight the viability of non-antibiotic adjunct therapies in optimizing antimicrobial regimens while mitigating the development of resistance. However, safety and toxicity evaluations of the fabricated nanocarrier in relevant ocular models are required to confirm its biocompatibility. Besides extensive testing against a broad range of clinical isolates of *P. aeruginosa* to address antimicrobial variability, future investigations should also seek to obtain mechanistic insights into N-acetylcysteine’s antimicrobial synergy. Ultimately, clinical trials are crucial in evaluating safety, efficacy, pharmacokinetics, and patient compliance, with the aim of translating this promising nanocarrier system into clinical use for the treatment of bacterial keratitis.

## Figures and Tables

**Figure 1 pharmaceutics-17-01184-f001:**
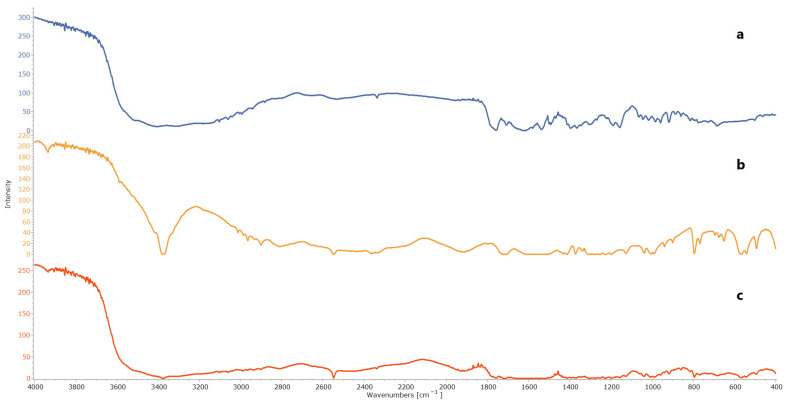
FTIR of (a) CTZ, (b) NAC, and (c) physical mixture of CTZ and NAC.

**Figure 2 pharmaceutics-17-01184-f002:**
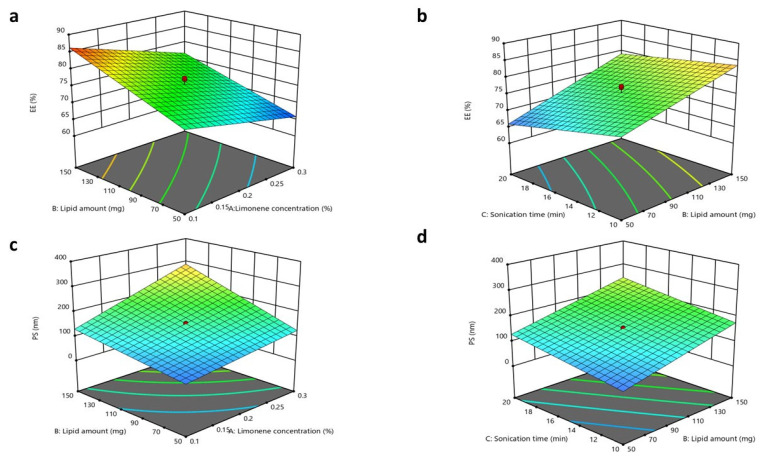
Response 3D plots for the influence of (X_1_) limonene concentration, (X_2_) lipid amount, and sonication time (X_3_) on (**a**,**b**) EE% and (**c**,**d**) PS.

**Figure 3 pharmaceutics-17-01184-f003:**
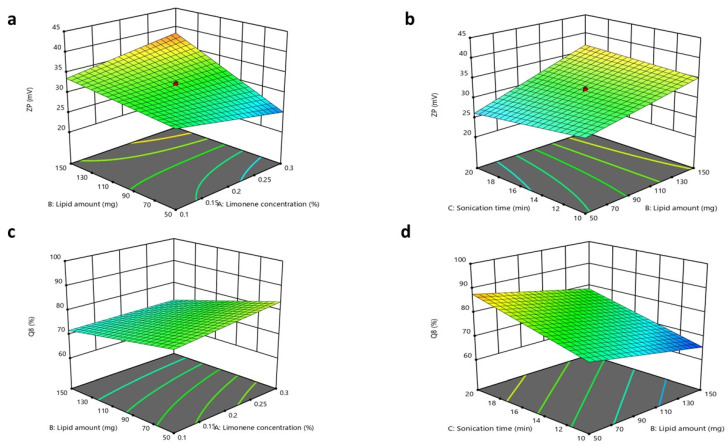
Response 3D plots for the impacts of (X_1_) limonene concentration, (X_2_) lipid amount, and sonication time (X_3_) on (**a**,**b**) ZP and (**c**,**d**) Q8.

**Figure 4 pharmaceutics-17-01184-f004:**
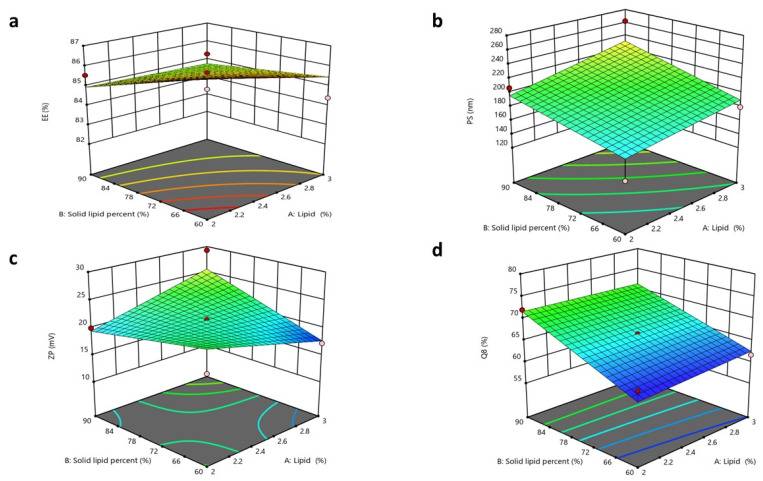
Response 3D plots for the impacts of (X_1_) lipid concentration and (X_2_) solid lipid percentage on (**a**) EE%, (**b**) PS, (**c**) ZP, (**d**) Q8.

**Figure 5 pharmaceutics-17-01184-f005:**
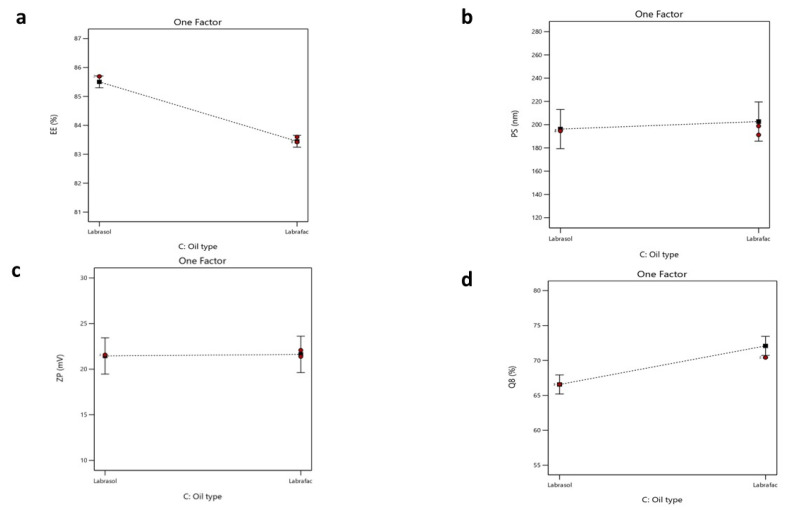
Single plot for the impact of (X_3_) oil type on (**a**) EE%, (**b**) PS, (**c**) ZP, and (**d**) Q8.

**Figure 6 pharmaceutics-17-01184-f006:**
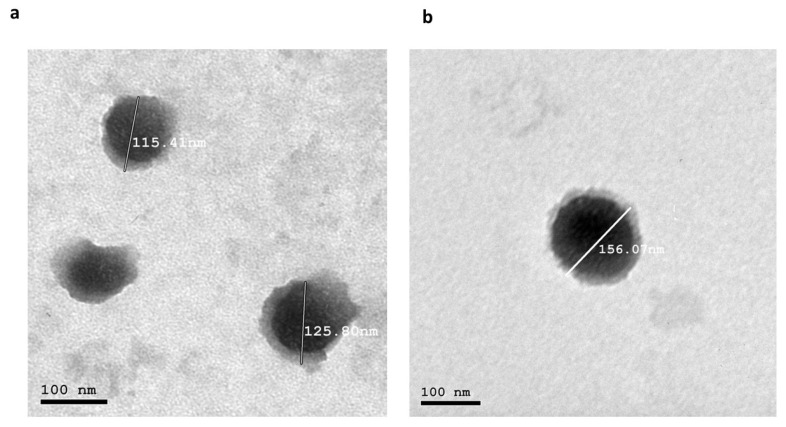
(**a**) Morphology of the optimized INV formula and (**b**) morphology of the optimized NLC. Abbreviations: INV, invasome; NLC, nanostructured lipid carrier.

**Figure 7 pharmaceutics-17-01184-f007:**
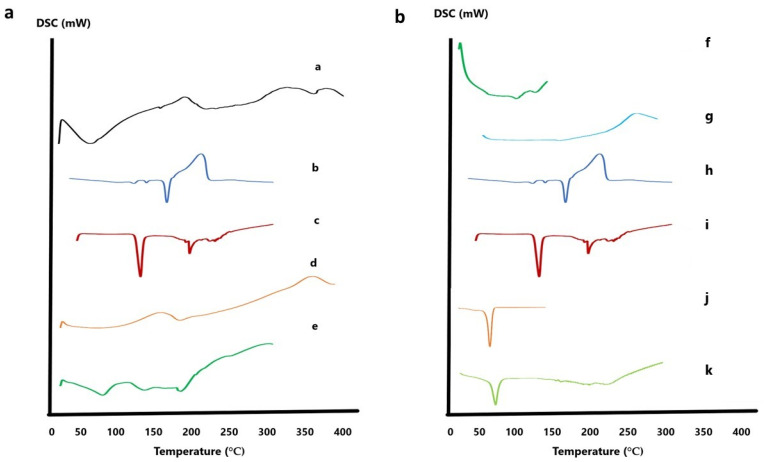
(**a**) DSC thermograms of optimized INV formula: (a) lipid, (b) CTZ, (c) NAC, (d) limonene, and (e) CTZ-loaded INV formula. (**b**) DSC thermograms of optimized NLC formula: (f) Tween 80, (g) Labrasol, (h) CTZ, (i) NAC, (j) Compritol, and (k) CTZ-loaded NLC formula. Abbreviations: DSC, differential scanning calorimetry; NAC, N-acetylcysteine; CTZ, ceftazidime; INV, invasome; NLC, nanostructured lipid carrier.

**Figure 8 pharmaceutics-17-01184-f008:**
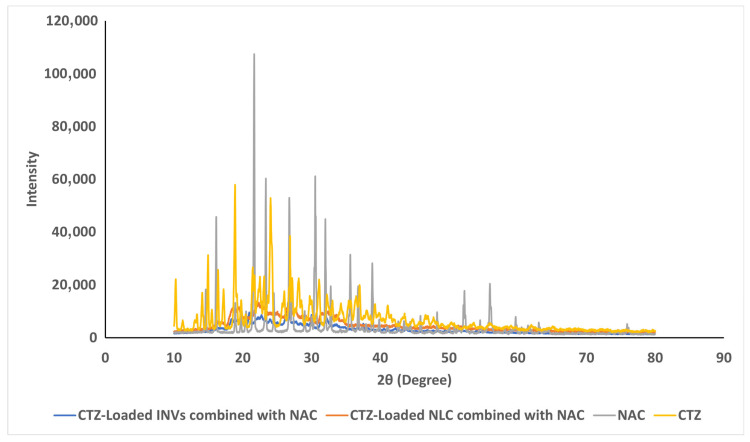
The XRPD curves of pure CTZ, NAC, and the lyophilized formulae.

**Figure 9 pharmaceutics-17-01184-f009:**
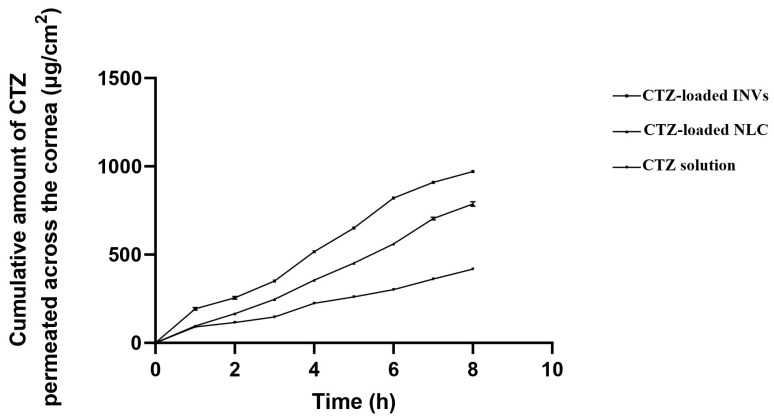
Ex vivo permeation profiles of different formulae.

**Figure 10 pharmaceutics-17-01184-f010:**
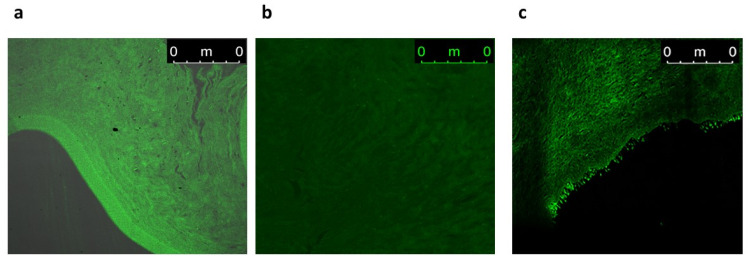
A tile scan CLSM photomicrograph of a longitudinal section in a cow cornea treated with (**a**) FDA-labeled INVs, (**b**) FDA-labeled NLCs, and (**c**) FDA aqueous solution.

**Figure 11 pharmaceutics-17-01184-f011:**
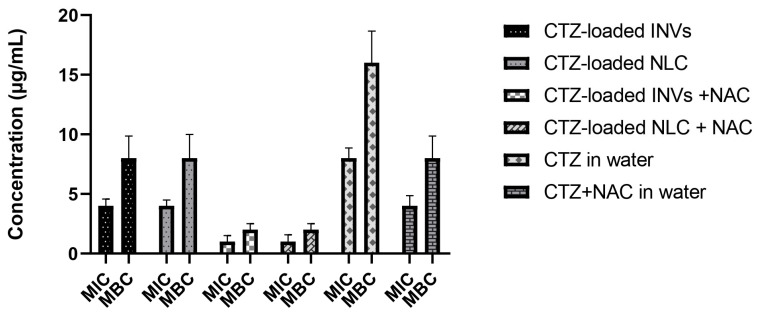
MICs and MBCs of the tested formulations.

**Figure 12 pharmaceutics-17-01184-f012:**
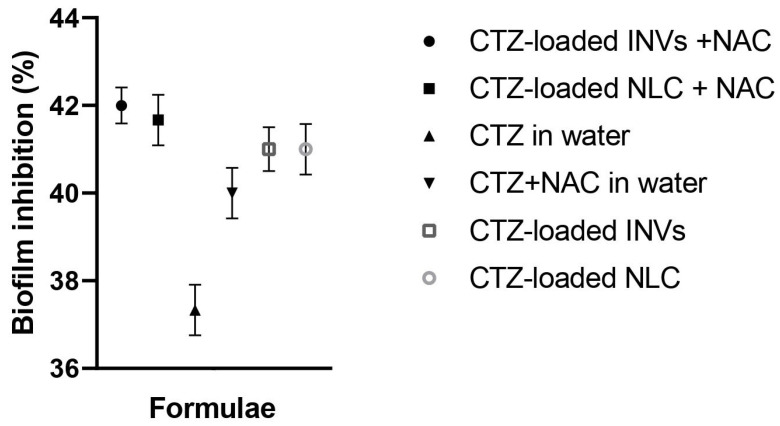
Effects of various sub-MIC values (0.5–0.25 MIC) of CTZ-loaded INVs, CTZ-loaded NLC, CTZ-loaded INVs combined with NAC, CTZ-loaded NLC combined with NAC, CTZ alone, and aqueous solution of CTZ and NAC on a clinical isolate from a *P. aeruginosa* biofilm. Results are expressed as the percentage of biofilm inhibition.

**Figure 13 pharmaceutics-17-01184-f013:**
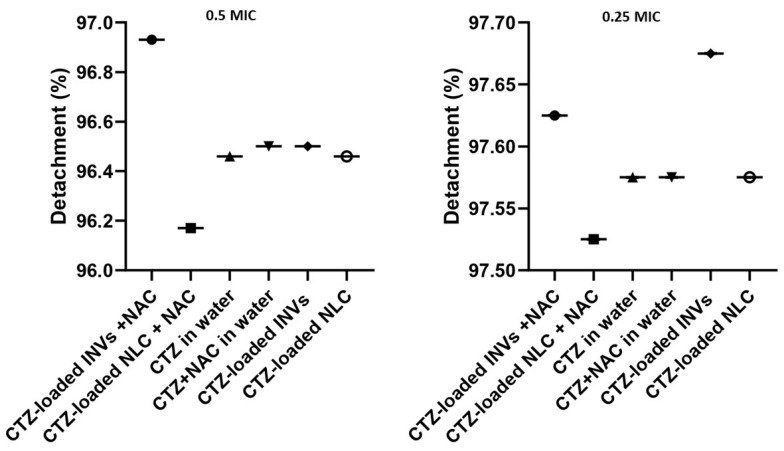
Biofilm detachment effects of CTZ-based formulae at sub-MIC levels against a pre-formed biofilm, measured at OD 570 nm.

**Figure 14 pharmaceutics-17-01184-f014:**
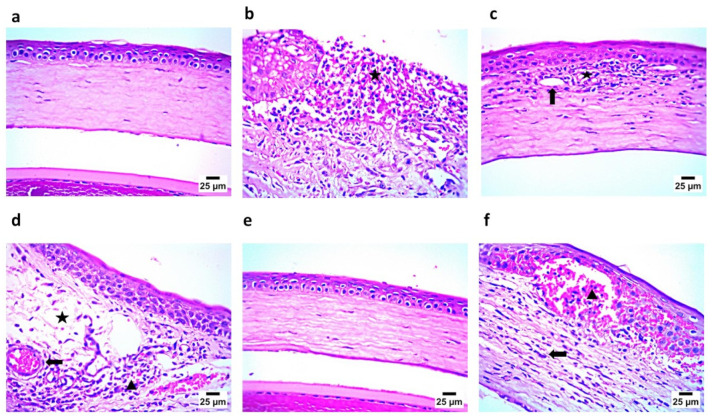
(**a**) Photomicrograph showing Gp1 with normal corneal histological structure, including epithelium and stroma; (**b**) photomicrograph showing Gp2 with complete sloughing of corneal epithelium and infiltration of stroma by a high number of inflammatory cells (star); (**c**) photomicrograph showing Gp3 with newly formed blood vessels (arrow) and infiltration of stroma by a small number of inflammatory cells (star); (**d**) photomicrograph showing Gp4 with mild stromal edema (star) and infiltration by a moderate number inflammatory cells (arrowhead), in addition to congestion of newly formed blood vessels (arrow); (**e**) photomicrograph showing Gp5 with normal corneal histological structure including epithelium and stroma; and (**f**) photomicrograph showing Gp6 with severe hemorrhage in corneal epithelium and stroma (arrowhead) and infiltration of stroma by a moderate number of inflammatory cells, mainly lymphocytes and eosinophils (arrow) (hematoxylin and eosin staining).

**Figure 15 pharmaceutics-17-01184-f015:**
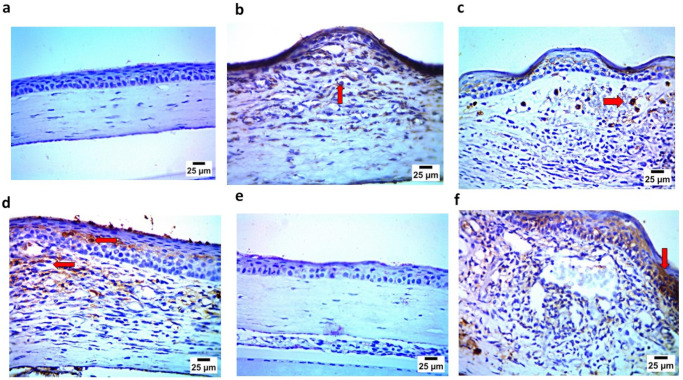
(**a**) Photomicrograph showing Gp1 with negative expression of caspase 3 in corneal epithelium; (**b**) photomicrograph showing Gp2 with positive expression of caspase 3 in stroma; (**c**) photomicrograph showing Gp3 with mild positive expression of caspase 3 in stroma (arrow); (**d**) photomicrograph showing Gp4 with mild positive expression of caspase 3 in corneal epithelium and in stroma (arrow); (**e**) photomicrograph showing Gp5 with negative expression of caspase 3 in corneal epithelium and stroma; and (**f**) photomicrograph showing Gp6 with strong positive expression of caspase 3 in corneal epithelium (arrow) (IHC-peroxidase-DAB).

**Figure 16 pharmaceutics-17-01184-f016:**
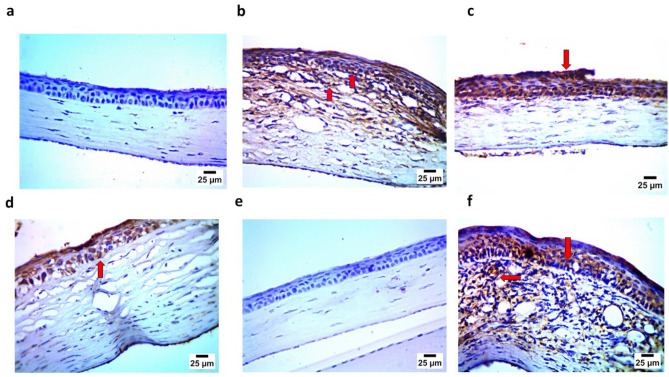
(**a**) Photomicrograph showing Gp1 with negative expression of IL1 β in corneal epithelium and stroma; (**b**) photomicrograph showing Gp2 with strong positive expression of IL1 β in corneal epithelium and stroma (arrow); (**c**) photomicrograph showing Gp3 with mild positive expression of IL1 β in corneal epithelium (arrow); (**d**) photomicrograph showing Gp4 with moderate positive expression of IL1 β in corneal epithelium (arrow); (**e**) photomicrograph showing Gp5 with negative expression of IL1 β in corneal epithelium and stroma; and (**f**) photomicrograph showing Gp6 with strong positive expression of IL1 β in corneal epithelium and stroma (arrow) (IHC–peroxidase–DAB).

**Table 1 pharmaceutics-17-01184-t001:** The 2^3^ factorial design for CTZ-loaded INVs.

Factor	Level	
	Low	High
X_1_: Limonene concentration (%)	0.1	0.3
X_2_: Lipid amount (mg)	50	150
X_3_: Sonication time (min)	10	20
Responses	Constraints	
Y_1_: EE (%)	Maximize	
Y_2_: PS (nm)	Minimize	
Y_3_: PDI	Minimize	
Y_4_: ZP (mV)	Maximize	
Y_5_: Q8 (%)	Maximize	

Abbreviations: EE%, entrapment efficiency percentage; CTZ, ceftazidime; PS, particle size; PDI, polydispersity index; ZP, zeta potential; Q8, percentage drug release after 8 h; INVs, invasomes.

**Table 2 pharmaceutics-17-01184-t002:** The 2^3^ factorial design for CTZ-loaded NLCs.

Factor	Level	
	Low	High
X_1_: Lipid concentration (%)	2	3
X_2_: Percentage of solid lipids (%)	60	90
X_3_: Oil type	Labrasol	Labrafac
Responses	Constraints	
Y_1_: EE (%)	Maximize	
Y_2_: PS (nm)	Minimize	
Y_3_: PDI	Minimize	
Y_4_: ZP (mV)	Maximize	
Y_5_: Q8 (%)	Maximize	

Abbreviations: EE%, entrapment efficiency percentage; CTZ, ceftazidime; PS, particle size; PDI, polydispersity index; ZP, zeta potential; Q8, percentage of drug released after 8 h; NLC, nanostructured lipid carrier.

**Table 3 pharmaceutics-17-01184-t003:** Output data of the 2^3^ factorial design analysis of INV formulations.

Source	EE (%)	PS (nm)	PDI	ZP (mV)	Q8 (%)
*p*-value	<0.0001	<0.0001	0.7771	<0.0001	0.0001
X_1_ = A = Limonene concentration	0.0005	<0.0001	0.4036	0.6572	0.1705
X_2_ = B = Lipid amount	<0.0001	<0.0001	0.7231	<0.0001	0.0009
X_3_ = C = Sonication time	0.0047	<0.0001	0.6690	0.2342	<0.0001
Adequate precision R^2^	20.8153 0.928	82.1787 0.9984	2.0730 0.1218	24.4246 0.9756	17.13 0.9116
Adjusted R^2^	0.901	0.9964	−0.2076	0.9552	0.8784
Predicted R^2^	0.7781	0.9438	−1.7956	0.7596	0.7421
Significant factors	X_1_, X_2_, X_3_	X_1_, X_2_, X_3_	–	X_2_	X_2_, X_3_
Predicted value of selected formula	87.34	123.36	0.493	33.40	64.93
Observed value of selected formula	87.08	114.50	0.493	33.38	65.66

**Table 4 pharmaceutics-17-01184-t004:** Output data of 2^3^ factorial design analysis of NLC formulations.

Source	EE (%)	PS (nm)	PDI	ZP (mV)	Q8 (%)
*p*-value	<0.0001	0.0007	0.0086	<0.0001	0.001
X_1_ = A = Lipid concentration	0.0056	0.0015	0.0954	0.1054	0.4816
X_2_ = B = Solid lipid percentage	0.0006	0.0006	0.4430	0.0001	0.0001
X_3_ = C = Oil type	<0.0001	0.1456	0.0017	0.3446	0.0008
Adequate precision R^2^	23.4792 0.987	24.9572 0.9755	11.0940 0.9320	57.4204 0.9971	16.8556 0.9722
Adjusted R^2^	0.9715	0.946	0.8504	0.9937	0.9389
Predicted R^2^	0.7942	0.812	−0.4163	0.9789	0.7402
Significant factors	X_1_, X_2_, X_3_	X_1_, X_2_	−	X_2_	X_2_, X_3_

Abbreviations: EE%, entrapment efficiency percentage; PS, particle size; PDI, polydispersity index; Q8%, cumulative percentage of drug permeated after 8 h.

## Data Availability

The datasets generated during this study are accessible from the corresponding author upon request.

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
