# Peer review of "Invasomes and Nanostructured Lipid Carriers for Targeted Delivery of Ceftazidime Combined with N-Acetylcysteine: A Novel Approach to Treat Pseudomonas aeruginosa-Induced Keratitis"

_pharmaceutics, 2025, doi:10.3390/pharmaceutics17091184_

Round 1
Reviewer 1 Report
Comments and Suggestions for Authors
The authors have clearly demonstrated the effect of INVs and NLC containing CTZ and NAC in corneal permeation and antibacterial activity. Experimental methods and results are well written, and the data support the claims of the authors. The following are some suggestions and questions which will further improve the quality of the paper.
- The synthesis and characterisation sections of INVs and NLC are essential, but they are taking up a lot of space in the main text. It would be beneficial to combine the figures wherever possible and keep the detailed tables in the supplementary information.
- Is it possible to compare a commercially available CTZ formulation to the current strategy?
- There are many reports in the literature showing the antibacterial effect of CTZ nanoparticles against Pseudomonas aeruginosa. What are the advantages of INVs and NLC over nanoparticles?
- There are cyclodextrin-mediated strategies to combine CTZ and vancomycin for making ophthalmic formulations for the treatment of keratitis. Can the current strategy be used to combine different antibiotics for keratitis treatment?
- Authors may include a separate discussion summarising the key findings, possible future directions, especially in clinical applications of their approach and potential limitations.
Author Response
The authors appreciate the reviewer's efforts in revising the manuscript entitled: " Invasomes and Nanostructured Lipid Carriers for Targeted Delivery of Ceftazidime Combined with N-Acetylcysteine: A Novel Approach to Treat Pseudomonas aeruginosa-Induced Keratitis ". Please find below our responses to the reviewer's comments.
Reviewer #1:
The authors thank the reviewer for the valuable comments and careful revision.
- The synthesis and characterisation sections of INVs and NLC are essential, but they are taking up a lot of space in the main text. It would be beneficial to combine the figures wherever possible and keep the detailed tables in the supplementary information.
Response: Thank you for your valuable comment. The sections concerning the characterization of the optimized formula were combined, and the DSC and TEM figures were consolidated. Detailed tables were added as supplementary material.
- Is it possible to compare a commercially available CTZ formulation to the current strategy?
Response: Thank you for your valuable comment. Currently, there are no widely commercially approved ocular products of CTZ due to its poor aqueous stability and rapid degradation in solution. Only compounded formulations prepared by pharmacists on demand. This section was added to the revised manuscript. Please revise lines 98-100.
- There are many reports in the literature showing the antibacterial effect of CTZ nanoparticles against Pseudomonas aeruginosa. What are the advantages of INVs and NLC over nanoparticles?
Response: Thank you for your valuable comment. The current study focuses on utilizing simple and scalable nanocarriers that exhibit low ocular toxicity, enhanced corneal penetration capabilities, and controlled release properties. Therefore, two nanocarriers were selected and compared to choose optimal nanocarrier for delivery CTZ: one is invasomes, which is lessly used in the ocular delivery, a deformable vesicular nanocarrier that may possess antimicrobial activity due to its terpenoid components, and the other is nanostructured lipid carriers (NLC), which have low toxicity, improved stability, provide sustained release, and commonly used in ocular delivery. Please revise lines 100-112
- There are cyclodextrin-mediated strategies to combine CTZ and vancomycin for making ophthalmic formulations for the treatment of keratitis. Can the current strategy be used to combine different antibiotics for keratitis treatment?
Response: Thank you for your valuable comment. Yes, the current strategy of combining an antibiotic-loaded nanocarrier with another free antibiotic for keratitis treatment is similar to how cyclodextrin-mediated strategies enable the combination of incompatible antibiotics, such as ceftazidime and vancomycin, in ophthalmic formulations. The free soluble antibiotic can be simply mixed or added to the finished nanocarrier suspension or gel, allowing it to act in conjunction with the nanoparticle-delivered antibiotic. This can avoid stability or drug loading challenges that might arise if the second antibiotic was encapsulated directly.
- Authors may include a separate discussion summarising the key findings, possible future directions, especially in clinical applications of their approach and potential limitations.
Response: Thank you for your valuable comment. Please revise lines 917-944.
Reviewer 2 Report
Comments and Suggestions for Authors
Thanks for the hard-work and the well structured manuscript.
I have some minor comments:
- A typo error in the first line of the abstract (noncarrier), I believe it should be nano carrier
- The FTIR in methodology needs more elaboration on the preparation method.
- It would be better if an PXRD study was included to study if there is a change in the crystallinity of the drug or amorphous formation (if possible).
- Is there any rationale for the dose of NAC of 75 mg/ml?
Author Response
The authors appreciate the reviewer's efforts in revising the manuscript entitled: " Invasomes and Nanostructured Lipid Carriers for Targeted Delivery of Ceftazidime Combined with N-Acetylcysteine: A Novel Approach to Treat Pseudomonas aeruginosa-Induced Keratitis ". Please find below our responses to the reviewer's comments.
Reviewer #2:
The authors thank the reviewer for the valuable comments and careful revision.
- A typo error in the first line of the abstract (noncarrier), I believe it should be nano carrier
Response: Thank you for your careful revision. The term has been corrected.
- The FTIR in methodology needs more elaboration on the preparation method.
Response: Thank you for your careful comment. Please revise section 2.2.
- It would be better if an PXRD study was included to study if there is a change in the crystallinity of the drug or amorphous formation (if possible).
Response: Thank you for your careful comment. Please revise sections 2.11.3& 3.7.3& Figure 8.
- Is there any rationale for the dose of NAC of 75 mg/ml?
Response: Thank you for your careful comment. Based on a previous study concerning the combination of NAC (75 mg/ml) with various antibiotics, including CTZ, to reduce the minimum biofilm inhibitory concentration of several gram-negative bacteria, including Pseudomonas aeruginosa. Additionally, primary screening was performed, where the MIC and MBC of the CTZ-loaded formulation and pure CTZ were evaluated in the presence of various NAC concentrations, starting from 1.6mg/ml (10 mM). A significant reduction in the minimum biofilm inhibitory concentration was observed at a concentration of 75 mg/mL, which was consistent with the previous study. The reference was added to Section 2.10 of the methodology.
Reviewer 3 Report
Comments and Suggestions for Authors
This manuscript presents a designed study. It can be recommended for publication after major revisions addressing the clarity of certain methodological choices, detailed comparison metrics, and a discussion of limitations. The manuscript contributes meaningfully to the field of targeted antimicrobial therapy and nanotechnology-enabled drug delivery.
- Invasomes and Nanostructured Lipid Carriers for Targeted Delivery of Ceftazidime Combined with N-Acetylcysteine: A Novel Approach to Treat Pseudomonas aeruginosa-Induced Keratitis.
- Suggested to incorporate more recent statistics or global epidemiological data on P. aeruginosa-induced keratitis, if available, to highlight clinical significance.
- More information on selection rationale for concentrations and types of lipids/terpenes/oils in formulations may help readers replicate or build upon the work.
- For antimicrobial and biofilm assays, clarify if clinical isolates vs. reference strains were always used, and how many replicate experiments were performed.
- The in vivo animal model section is thorough; consider including more details on randomization, blinding, or humane endpoints for animal studies.
- The manuscript demonstrates that INVs outperformed NLCs in corneal permeation and deposition, while both offered synergistic antimicrobial activity when combined with NAC. Thus, Include numerical values for key comparative parameters in the text for quick reference.
- Discuss potential limitations such as variability in clinical isolates, potential toxicity/irritation of formulations, or translation challenges to human subjects.
- Suggest highlighting the proposed next steps for translational/clinical evaluation or addressing any identified limitations in conclusions sections
- There are some minor grammatical corrections and sentence structure streamlining in several sections will improve readability.
Author Response
The authors appreciate the reviewer's efforts in revising the manuscript entitled: " Invasomes and Nanostructured Lipid Carriers for Targeted Delivery of Ceftazidime Combined with N-Acetylcysteine: A Novel Approach to Treat Pseudomonas aeruginosa-Induced Keratitis ". Please find below our responses to the reviewer's comments.
Reviewer #3:
The authors thank the reviewer for the valuable comments and careful revision.
- Suggested to incorporate more recent statistics or global epidemiological data on P. aeruginosa-induced keratitis, if available, to highlight clinical significance.
Response: Thank you for your careful comment. Please revise lines 66-71.
- More information on selection rationale for concentrations and types of lipids/terpenes/oils in formulations may help readers replicate or build upon the work.
Response: Thank you for your careful comment. Please revise sections 3.2& 3.4.
- For antimicrobial and biofilm assays, clarify if clinical isolates vs. reference strains were always used, and how many replicate experiments were performed.
Response: Thank you for your careful comment. All assays were performed on clinical isolates (no reference strains were included). Each experiment was conducted in three biological replicates (independent cultures started on different days), with each measured in triplicate technical wells (n = 3 × 3 per condition).
- The in vivo animal model section is thorough; consider including more details on randomization, blinding, or humane endpoints for animal studies.
Response: Thank you for your careful comment. Please revise section 2.14
- The manuscript demonstrates that INVs outperformed NLCs in corneal permeation and deposition, while both offered synergistic antimicrobial activity when combined with NAC. Thus, Include numerical values for key comparative parameters in the text for quick reference.
Response: Thank you for your careful comment. Please revise section 3.12
- Discuss potential limitations such as variability in clinical isolates, potential toxicity/irritation of formulations, or translation challenges to human subjects.
Response: Thank you for your careful comment. Please revise lines 929-944
- Suggest highlighting the proposed next steps for translational/clinical evaluation or addressing any identified limitations in conclusions sections
Response: Thank you for your careful comment. Please revise section 4
- There are some minor grammatical corrections and sentence structure streamlining in several sections will improve readability.
Response: Thank you for your careful comment.The manuscript was revised using the premium version of Grammarly.
Round 2
Reviewer 3 Report
Comments and Suggestions for Authors
The authors have reflected all the said suggestions and comments, which made the manuscript enhanced with improved readability; Thus, I suggest for further consideration with acceptance.